# Plasticity Activation via Polar Operator: A Plug-in Method for Balancing Stability and Plasticity

**Guodong Zheng** [1]  **Enneng Yang** [2]  **Xiaoyan Wang** [3]  **Yihan Chen** [4]  **Feihong He** [2]  **Quan Zheng** [1]  **Peng Wang** [2]  **Li Shen** [2]

## Abstract

Continual learning (CL) seeks models that acquire new knowledge while avoiding catastrophic forgetting. However, many methods that mitigate forgetting constrain parameter updates and thereby reduce model plasticity. We revisit the singular value spectrum of gradients in representative CL methods and show that they commonly exhibit singular value collapse, where only a small subset of gradient directions drive parameter updates. Motivated by this observation, we propose **P**lasticity **A**ctivation via **P**olar **O**perator (PAPO), a plug-in that preserves the dominant directions that mitigate forgetting while activating previously suppressed directions to enhance plasticity. Concretely, PAPO modifies the gradient $\mathbf{G}$ as $\mathbf{G} \leftarrow \mathbf{G} + \lambda \cdot \mathrm{polar}(\mathbf{G})$, which uniformly increases near-zero singular values without changing the singular vectors. To avoid the cost of explicit singular value decomposition, we approximate the polar factor using the iteration-dependent Polar Express scheme, which relies only on matrix multiplications and additions. In our empirical evaluation on both vision and language benchmarks, incorporating PAPO yields consistent improvements. In particular, on Mini-ImageNet, integrating PAPO into ER, MAS, GPM and TRGP produces substantial accuracy gains of 9.01%, 4.76%, 8.90% and 9.19%, respectively.

[1]Huazhong University of Science and Technology, Wuhan, China [2]Shenzhen Campus of Sun Yat-sen University, China [3]Information Technology Service Center of People's Court, Beijing, China [4]University of Science and Technology of China, Hefei, China. Correspondence to: Li Shen <mathshenli@gmail.com>, Peng Wang <wangp389@mail.sysu.edu.cn>, Xiaoyan Wang <428163395@139.com>.

*Proceedings of the 43rd International Conference on Machine Learning*, Seoul, South Korea. PMLR 306, 2026. Copyright 2026 by the author(s).

## 1. Introduction

Continual learning (CL) aims to enable models to acquire new knowledge as tasks arrive while retaining previously acquired knowledge. Prior work has focused primarily on mitigating catastrophic forgetting (McCloskey & Cohen, 1989), often by constraining parameter updates. However, overly restricting updates to prevent forgetting can impair the model's ability to learn new tasks. This tension, commonly referred to as the stability-plasticity dilemma (Grossberg, 2013; Wang et al., 2024b), captures the inherent difficulty of preserving past knowledge while accommodating new information.

Recent work has sought to balance the stability–plasticity trade-off using diverse strategies. Examples include auxiliary-network approaches that separate stability and plasticity roles (Kim et al., 2023; Lu et al., 2025a), methods tailored to particular CL algorithms such as null-space approaches (Kong et al., 2022) and model merging (Yang et al., 2026), and methods designed for specific CL paradigms such as continual meta-learning (Chen et al., 2023) and streaming learning (Elsayed & Mahmood, 2024). Many of these approaches either require additional network components or are tailored to specific algorithms or learning scenarios, and therefore do not provide a broadly applicable and efficient solution to the stability–plasticity dilemma.

To develop a general method for enhancing plasticity in CL approaches that mitigate forgetting, we examine the gradient singular value spectra of three representative methods: Experience Replay (ER) (Chaudhry et al., 2019b), Memory Aware Synapses (MAS) (Aljundi et al., 2018), and Gradient Projection Memory (GPM) (Saha et al., 2021). By inspecting these spectra, we identify a common pattern, which we call *singular value collapse*: a small set of dominant singular values is followed by a long tail of near-zero singular values. Under singular value collapse, only a small subset of gradient directions drives parameter updates, thereby constraining model plasticity. Specifically, the spectrum of GPM exhibits a cliff that separates a few relatively large singular values from a bulk of near-zero values, and this cliff shifts leftward as tasks accumulate. We also provide a theoretical analysis showing that this phenomenon naturally

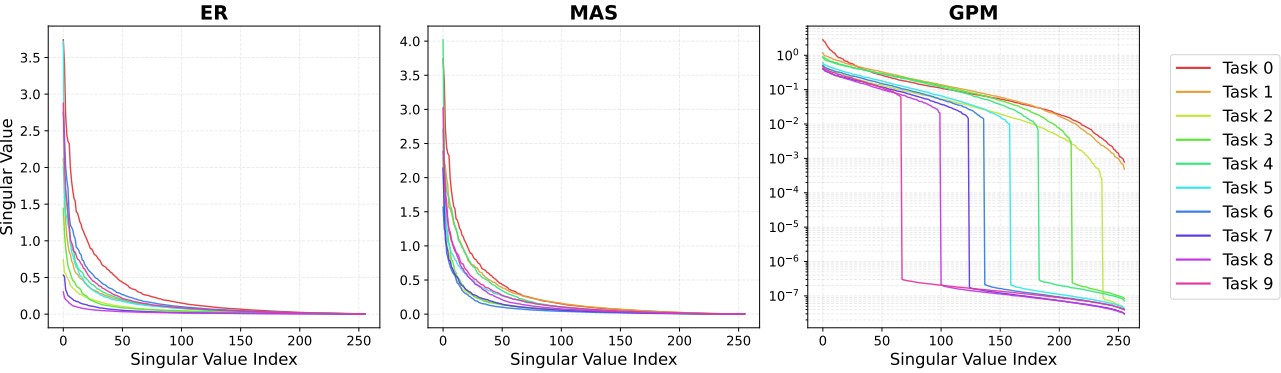

*Figure 1.* Gradient singular value spectra of a convolutional layer of AlexNet on CIFAR-100 for ER, MAS, and GPM across tasks.

arises in orthogonal-projection-based methods.

Motivated by the observed singular value collapse, we propose Plasticity Activation via Polar Operator (PAPO), a plug-in module that activates previously suppressed singular directions to enhance plasticity in CL methods. Specifically, PAPO modifies the gradient as $\mathbf{G} \leftarrow \mathbf{G} + \lambda \cdot \mathrm{polar}(\mathbf{G})$, where $\mathrm{polar}(\cdot)$ denotes the polar operator and $\lambda > 0$ is a balance coefficient. Applying the polar operator preserves the singular vectors while shifting every singular value $\sigma_i$ to $\sigma_i + \lambda$. As a result, the relative ordering of singular values is maintained and the original dominant update directions are retained, while previously near-zero singular values are uniformly increased and the corresponding directions are activated. To avoid an explicit singular value decomposition (SVD), we approximate $\mathrm{polar}(\cdot)$ with Polar Express (Amsel et al., 2025), an iterative scheme that uses only matrix multiplications and additions and is therefore efficient on modern GPUs.

Empirically, we conduct experiments on both vision and language tasks. For vision tasks, we integrate PAPO into ER, MAS, GPM and Trust Region Gradient Projection (TRGP) (Lin et al., 2022) and evaluate the resulting methods on four standard benchmarks: PMNIST, CIFAR-100, MiniImageNet and the 5-Datasets sequence. Averaged across these benchmarks, PAPO improves the accuracy of ER, MAS, GPM and TRGP by 3.52%, 3.18%, 2.80% and 2.91%, respectively. On MiniImageNet in particular, PAPO yields substantial accuracy gains of 9.01%, 4.76%, 8.90% and 9.19%, while also improving backward transfer (BWT) by 0.03, 0.04, 0.04 and 0.01, respectively. For language tasks, we apply PAPO to InfLoRA (Liang & Li, 2024) and GainLoRA (Liang et al., 2026) and evaluate on the SuperNI benchmark (Wang et al., 2022b). PAPO improves the accuracy of InfLoRA and GainLoRA by 3.37% and 1.24%, respectively, with corresponding reductions in forgetting of 3.22% and 1.39%. These results demonstrate the practical benefits of PAPO as a plug-in that can be incorporated into methods that mitigate forgetting in CL. Our code is available

at https://github.com/Zhenggd943/PAPO.

Our main contributions are summarized as follows:

- We revisit the singular value spectrum of gradients in ER, MAS and GPM and identify a phenomenon we term *singular value collapse*. We also provide a theoretical analysis that explains why a spectral "cliff" arises in orthogonal-projection-based methods.

- We propose a plug-in, Plasticity Activation via Polar Operator (PAPO), which activates suppressed gradient directions in the optimization dynamics while preserving the original mitigating-forgetting directions. PAPO is broadly applicable and does not require changes to model architecture or loss functions.

- We validate PAPO through extensive experiments on both vision and language benchmarks, verifying that it consistently improves average performance across these tasks.

**Related Work.** Due to space constraints, a comprehensive literature review is provided in Appendix A.

## 2. Preliminary

In this section, we introduce the notation used throughout the paper and provide a overview of the SVD of matrices.

**Continual Learning.** We consider the standard CL setting with $T$ sequential training tasks $\{\mathcal{D}_1, \dots, \mathcal{D}_T\}$, where $\mathcal{D}_\tau = \{(\mathbf{X}_\tau, \mathbf{Y}_\tau)\}$ denotes the training dataset of task $\tau$. For each task $\tau \in \{1, \dots, T\}$, we denote the network parameters by $\{\mathbf{W}_{\tau,l}\}_{l=1}^L$, where $\mathbf{W}_{\tau,l}$ represents the weight matrix of the $l$-th layer. For notational simplicity, we omit the layer index when it is clear from the context and write $\mathbf{W}_\tau$ to denote the network weights of task $\tau$.

**Singular value decomposition.** Let $\mathbf{W} \in \mathbb{R}^{n \times d}$ be a

matrix of rank $r \leq \min(n, d)$. Its SVD can be written as:

$$\mathbf{W} = \mathbf{U}\Sigma\mathbf{V}^\top = \sum_{i=1}^{r} \sigma_i \mathbf{u}_i \mathbf{v}_i^\top, \qquad (1)$$

where $\mathbf{U} = [\mathbf{u}_1, \ldots, \mathbf{u}_r] \in \mathbb{R}^{n \times r}$ and $\mathbf{V} = [\mathbf{v}_1, \ldots, \mathbf{v}_r] \in \mathbb{R}^{d \times r}$ have orthonormal columns, and $\Sigma = \mathrm{diag}(\sigma_1, \ldots, \sigma_r) \in \mathbb{R}^{r \times r}$ contains the nonzero singular values. Throughout this work, we assume the singular values are ordered as $\sigma_1 \geq \sigma_2 \geq \cdots \geq \sigma_r > 0$. We denote by $\sigma_i(\mathbf{W})$ the $i$-th singular value of $\mathbf{W}$.

**SVD on convolutional filters.** While SVD is naturally defined for 2D matrices, a convolutional filter is typically represented as a 4D tensor, which makes direct decomposition inapplicable. To address this, we reshape the tensor into a matrix before applying SVD. Specifically, for a convolutional layer with $C_i$ input channels, $C_o$ output channels, and a $K \times K$ kernel, the filter tensor $\mathcal{W} \in \mathbb{R}^{C_o \times C_i \times K \times K}$ is reshaped into a matrix $\mathbf{W} \in \mathbb{R}^{(C_i K^2) \times C_o}$.

## 3. Revisiting Gradients in Continual Learning

In this section, we revisit the singular value spectrum of gradients in three canonical classes of CL methods: ER, MAS, and GPM.

### 3.1. Singular Value Spectrum of Gradients

Examining gradients provides direct insight into optimization dynamics. Recently, Lewandowski et al. (2025) link rank reduction of the gradient matrix to a loss of plasticity, showing that reduced rank limits the diversity of gradient directions available for parameter updates. However, Lewandowski et al. (2025) focus primarily on methods that enhance plasticity and did not analyze the gradient properties induced by methods that mitigate forgetting. In this paper, we bridge that gap by directly examining the singular value spectrum of gradients, which offers a geometrically interpretable and numerically stable characterization of gradient structure. Below, we focus on three canonical classes of CL methods, ER, MAS and GPM, to investigate the limitations that hinder further progress.

We conduct experiments with ER, MAS and GPM on the CIFAR-100 benchmark using an AlexNet backbone to revisit the singular value spectrum of the gradient across tasks. In Figure 1, the spectra for ER and MAS exhibit a decay: a small set of dominant singular values is followed by a long tail of near-zero singular values. We observe a special case in GPM: the spectrum displays a pronounced "cliff" that separates a few relatively large singular values from a bulk of near-zero values, and this cliff appears at smaller singular value indices as the number of tasks increases. We refer to these behaviors as *singular value collapse*: only a

small subset of singular values remains substantial, while the remainder are comparatively small and often close to zero. The singular value spectrum of GPM differs from the trend in ER and MAS because GPM directly projects gradients to remove components overlapping with previous tasks, whereas ER and MAS implicitly shape the gradient's spectral distribution. Additional plots are available in Appendix F.1. We will present a theoretical analysis in Section 3.2 that characterizes this phenomenon in GPM and generalizes it to orthogonal-projection-based methods.

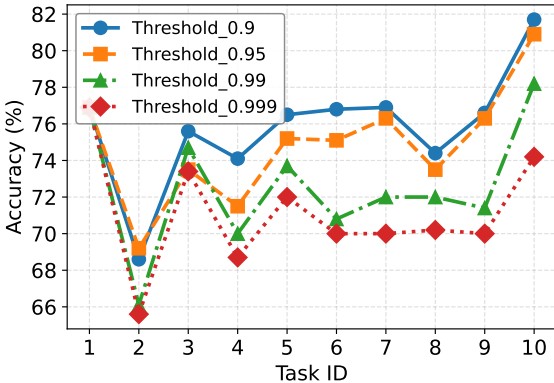

*Figure 2.* Accuracy curves under different GPM thresholds on the CIFAR-100 benchmark.

Next, we formally characterize singular value collapse. For any training step $t$ on task $\tau$ under an arbitrary continual-learning method, denote the gradient by $\mathbf{G}_\tau^t$. Without loss of generality, let $\mathrm{rank}(\mathbf{G}_\tau^t) = k$. Then, the singular value decomposition of $\mathbf{G}_\tau^t$ is

$$\mathbf{G}_\tau^t = \mathbf{U}_\tau^t \Sigma_\tau^t \mathbf{V}_\tau^{t\top} = \sum_{i=1}^{k} \sigma_i \mathbf{u}_i \mathbf{v}_i^\top. \qquad (2)$$

Under singular value collapse, a small subset of singular values is substantially larger than the remainder. Without loss of generality, suppose $\Sigma_\tau^t$ contains $r$ dominant singular values $\sigma_1, \ldots, \sigma_r$ while the remaining $k - r$ singular values $\sigma_{r+1}, \ldots, \sigma_k$ are comparatively small. From Equation (2) we therefore decompose the gradient $\mathbf{G}_\tau^t$ as

$$\mathbf{G}_\tau^t = \sum_{i=1}^{r} \sigma_i \mathbf{u}_i \mathbf{v}_i^\top + \sum_{j=r+1}^{k} \sigma_j \mathbf{u}_j \mathbf{v}_j^\top. \qquad (3)$$

As observed in Figure 1, a long tail of near-zero singular values frequently appears. Although the associated singular vectors are present, their contributions are strongly suppressed by these near-zero singular values. As a result, the second term becomes negligible in practice, leaving the first term to dominate the gradient direction. Consequently, the parameter update components responsible for mitigating

forgetting depend primarily on the leading singular vectors $\mathbf{u}_1, \ldots, \mathbf{u}_r$ and $\mathbf{v}_1, \ldots, \mathbf{v}_r$. We refer to these as the *mitigating-forgetting directions*. In these methods, parameter updates are thus dominated by these directions, while the remaining directions contribute only marginally.

We have established that singular value collapse effectively suppresses a subset of parameter update directions. To connect this phenomenon with a loss of plasticity, we substantiate the impairment by presenting a concrete example. Because GPM operates directly on gradients, it can more readily control the degree of singular value collapse. In practice, GPM uses a projection threshold to determine the number of orthogonal basis vectors retained for gradient projection. Appendix F.2 provides the singular value plots for different threshold settings. We find that a larger threshold increases the number of retained basis vectors and thus intensifies singular value collapse. This intensification produces a greater number of near-zero singular values in the tail. As shown in Figure 2, the accuracy curve shifts downward as the threshold increases, which indicates that *more severe singular value collapse exacerbates the loss of plasticity*.

**Discussions.** Singular value analysis in CL has been studied in prior work, and here we clarify how our approach differs. Zhu et al. (2021) perform an eigen-decomposition of the old and new feature extractors and define the angle between the corresponding eigenvectors to capture representation shifts during incremental learning. Guo et al. (2022) compute the eigenvalues and eigenvectors of the learned representations to compare the overall representation quality of two different representation learning methods.

## 3.2. Singular Value Collapse Analysis

As discussed in Section 3.1, the singular value spectrum exhibits a cliff that separates a few relatively large singular values from a bulk of near-zero values, and this cliff shifts leftward as tasks accumulate. We generalize this phenomenon to orthogonal-projection-based methods and provide a theoretical analysis that explains its emergence.

Orthogonal-projection-based methods have achieved notable success in mitigating forgetting by updating the weight $\mathbf{W}_\tau$ for task $\tau$ along directions orthogonal to the subspace $\mathcal{X}_\tau$ spanned by the previous $\tau - 1$ tasks (Farajtabar et al., 2020; Saha et al., 2021; Lin et al., 2022; Yang et al., 2023a). Motivated by the theoretical analysis of Zheng et al. (2026), we present a unified formulation of orthogonal projection based methods.

For any $\tau \in \{1, \ldots, T\}$, let $\mathcal{X}_\tau$ denote the subspace spanned by an orthonormal basis

$$\mathcal{B}_\tau = \{\mathbf{b}_{1,1}, \ldots, \mathbf{b}_{1,n_1}, \ldots, \mathbf{b}_{\tau-1,1}, \ldots, \mathbf{b}_{\tau-1,n_{\tau-1}}\}.$$

The subspace $\mathcal{X}_{\tau+1}$ is obtained by augmenting $\mathcal{X}_\tau$ with a set of new orthonormal vectors $\{\mathbf{b}_{\tau,1}, \ldots, \mathbf{b}_{\tau,n_\tau}\}$, so that

$$\mathcal{X}_{\tau+1} = \mathrm{span}\left(\mathcal{B}_\tau \cup \{\mathbf{b}_{\tau,1}, \ldots, \mathbf{b}_{\tau,n_\tau}\}\right).$$

The procedure used to select the new orthonormal vectors depends on the specific orthogonal-projection method. Saha et al. (2021) obtain these vectors as the leading left singular vectors of the per-layer representation matrix computed after each task. In contrast, Farajtabar et al. (2020) store a set of model gradient vectors and constructs an orthonormal basis via the Gram–Schmidt procedure.

Let $P_{\mathcal{X}_\tau}$ denote the orthogonal projector onto the subspace $\mathcal{X}_\tau$ and let $P_{\mathcal{X}_\tau}^\perp$ denote the projector onto its orthogonal complement. The gradient $\mathbf{G}_\tau^t$ at training step $t$ for task $\tau$ is then replaced by its projection onto the orthogonal complement,

$$\widetilde{\mathbf{G}}_\tau^t \;=\; P_{\mathcal{X}_\tau}^\perp\left(\mathbf{G}_\tau^t\right),$$

and $\widetilde{\mathbf{G}}_\tau^t$ is used for the subsequent parameter update.

Building on the unified formulation for orthogonal-projection-based methods established above, we provide a theoretical analysis of the singular value cliff phenomenon described in Section 3.1. The proof of Theorem 3.1 is deferred to Appendix C.

**Theorem 3.1.** *Let $\mathbf{G} \in \mathbb{R}^{n \times d}$ be a matrix, let $\mathcal{X}$ be an $m$-dimensional subspace of $\mathbb{R}^n$ spanned by the orthogonal basis $\{\mathbf{b}_1, \ldots, \mathbf{b}_m\}$, and let $P_{\mathcal{X}}^\perp$ be the orthogonal projection operator onto the orthogonal complement of $\mathcal{X}$. Then, the inequality $rank(P_{\mathcal{X}}^\perp(\mathbf{G})) \leq rank(\mathbf{G})$ holds. Moreover, if the intersection of the column space of $\mathbf{G}$ and $\mathcal{X}$ is non-trivial, then the rank strictly decreases, i.e., $rank(P_{\mathcal{X}}^\perp(\mathbf{G})) < rank(\mathbf{G})$.*

*Remark* 3.2. Theorem 3.1 shows that projecting a gradient matrix onto the orthogonal complement of a subspace yields a matrix whose rank is non-increasing. Moreover, if the column space of the gradient has a nontrivial intersection with the projected subspace, the rank decreases strictly. In orthogonal-gradient-based methods, the constraint subspace typically expands as training progresses. This expansion increases the probability of such intersections and hence produces progressive rank reduction. In particular, when rank reduction occurs, a tail of singular values becomes exactly zero, and the number of zero singular values tends to grow as tasks accumulate. This mechanism explains why the spectral cliff begins to appear at task 2 and why the cliff shifts leftward as the number of tasks increases in GPM (see Figure 1). In the extreme case where the constraint subspace spans the entire parameter space, the orthogonal projector $P_{\mathcal{X}_n}^\perp$ annihilates every matrix, so $\mathrm{rank}\left(P_{\mathcal{X}_n}^\perp(\mathbf{G})\right) = 0$. Consequently, the gradient vanishes and the model loses plasticity entirely.

# 4. Methodology

In this section, we introduce our method PAPO, which mitigates the negative impact of singular value collapse as discussed in Section 4.1. We also present efficient computational strategies for its components in Sections 4.2 and 4.3.

## 4.1. How can stability and plasticity be improved in CL?

As discussed in Section 3, most CL methods that mitigate forgetting suffer from singular value collapse, whereby a large fraction of singular directions contribute little or nothing to parameter updates. This phenomenon impairs model plasticity. Our objective is therefore to activate the remaining singular directions during optimization while preserving the dominant influence of the mitigating-forgetting directions. Accordingly, we pose the following question:

> **Q:** How can the mitigating-forgetting directions retain their dominant role in updates while the remaining singular directions are activated?

To address **Q**, we adopt two design desiderata that constrain how the gradient may be modified:

1. The singular vectors $\{\mathbf{u}_i\}_{i=1}^k$ and $\{\mathbf{v}_i\}_{i=1}^k$ remain unchanged after the modification.

2. The singular values associated with the mitigating-forgetting directions should remain larger than those of the remaining directions after the modification.

*Remark* 4.1. Condition 1 is a conservative design choice intended to preserve the original update *directions*. The singular vectors $\{\mathbf{u}_i\}$ and $\{\mathbf{v}_i\}$ define the principal left/right directions along which parameter updates act. Altering these vectors effectively changes the update basis and can therefore modify which components of previous tasks are reinforced or suppressed. Because our goal is to *activate* previously suppressed directions while retaining the mechanisms that mitigate forgetting, preserving the singular vectors prevents unintended perturbations of prior information.

*Remark* 4.2. Condition 2 guarantees that the modification does not demote the mitigating-forgetting directions. Concretely, if the post-modification singular values corresponding to the mitigating directions remain larger than those of the previously suppressed directions, the original dominant directions will continue to carry the larger update coefficients. This preserves stability while increasing plasticity by lifting near-zero singular values of the residual directions.

Motivated by Remark 4.1 and Remark 4.2, we introduce the polar operator as follows.

**Definition 4.3** (Polar operator). For any matrix **M** with the singular value decomposition $\mathbf{M} = \mathbf{U}\Sigma\mathbf{V}^\top$, the polar operator of **M** is defined as

$$\mathrm{polar}(\mathbf{M}) := \mathbf{U}\mathbf{V}^\top.$$

In fact, $\mathrm{polar}(\mathbf{M})$ denotes the closest semi-orthogonal matrix to **M** in the Frobenius norm sense (Higham, 2008). In particular, if **M** is square, then $\mathrm{polar}(\mathbf{M})$ is the orthogonal factor in the polar decomposition of **M**.

Based on the polar operator, we can address **Q** as follows:

$$\mathbf{G}_\tau^t = \mathbf{G}_\tau^t + \lambda \cdot \mathrm{polar}(\mathbf{G}_\tau^t) = \mathbf{U}_\tau^t \left(\Sigma_\tau^t + \lambda\mathbf{I}\right) \mathbf{V}_\tau^{t\top}, \quad (4)$$

where $\lambda$ is a balancing coefficient. Under singular value collapse, we can rewrite Equation (4) as follows:

$$\mathbf{G}_\tau^t = \underbrace{\sum_{i=1}^r (\sigma_i + \lambda) u_i v_i^\top}_{\text{Mitigate Forgetting}} + \underbrace{\sum_{j=r+1}^k (\sigma_j + \lambda) u_j v_j^\top}_{\text{Improve Plasticity}} \quad (5)$$

As shown in Equation (5), applying the polar operator increases all singular values by a uniform magnitude $\lambda$. Consequently, the relative ordering of the singular values remains invariant compared to the original gradient. Crucially, this operation preserves the singular vectors, thereby maintaining the original update directions. Hence, **Q** is addressed by incorporating the polar operator: it activates previously neglected gradient directions without disrupting the primary directions responsible for mitigating forgetting.

However, computing $\mathrm{polar}(\mathbf{G}_\tau^t)$ directly requires an SVD, which is computationally expensive if performed at every optimization step. In Section 4.2 we introduce an iterative algorithm that avoids SVD by approximating the polar factor using only matrix multiplications and additions. This is therefore well suited to efficient GPU implementation.

## 4.2. Computing the Polar Factor

Although $\mathrm{polar}(\mathbf{G}_\tau^t)$ can be obtained directly from the SVD, standard SVD algorithms are prohibitively expensive for deep-learning applications and do not fully exploit the parallelism available on GPUs (Amsel et al., 2025). Numerous numerical schemes have therefore been developed to compute the polar factor, including Newton–Schulz iterations (Higham, 2008), Padé iterations (Higham, 1986), QWHD iterations (Nakatsukasa et al., 2010), and Zolotarev-based polar decompositions (Nakatsukasa & Freund, 2016). Many of these methods require explicit matrix inverses or QR decompositions, operations that limit efficient GPU acceleration. By contrast, the Newton–Schulz approach is based on a polynomial approximation of the matrix sign function and uses only matrix multiplications and additions. Therefore, it is particularly well suited to GPU implementation.

**Newton–Schulz iterations.** Set the initial iterate as $\mathbf{X}_0 = \mathbf{G}_\tau^t / \|\mathbf{G}_\tau^t\|_{\mathrm{F}}$. For $k \geq 1$, the iterate $\mathbf{X}_k$ is updated as

$$\mathbf{X}_k = a\mathbf{X}_{k-1} + b\big(\mathbf{X}_{k-1}\mathbf{X}_{k-1}^\top\big)\mathbf{X}_{k-1} \qquad (6)$$
$$+ c\big(\mathbf{X}_{k-1}\mathbf{X}_{k-1}^\top\big)^2\mathbf{X}_{k-1},$$

where $a$, $b$ and $c$ are scalar coefficients. Convergence of the iteration in Equation (6) is promoted by choosing the coefficients so that the scalar polynomial

$$f(x) = ax + bx^3 + cx^5$$

has a fixed point at $x = 1$ with favorable local contraction properties.

**Jordan method.** Jordan et al. (2024) use a heuristic numerical search to propose a fixed-point iteration with coefficients $a = 3.4445$, $b = -4.7750$, and $c = 2.0315$. However, this iteration does not fully converge to $\mathrm{polar}(\mathbf{G}_\tau^t)$ and instead plateaus at an approximation error of approximately 0.3. Nevertheless, this method has been successfully employed in the MUON optimizer (Jordan et al., 2024).

Most methods based on the Newton–Schulz iteration employ fixed coefficients $a$, $b$, and $c$ at each step. In practice, this choice often leads to slow convergence in the early iterations, particularly when the initial iterate $\mathbf{X}_0$ is far from $\mathrm{polar}(\mathbf{G}_\tau^t)$. To address this issue, Polar Express (Amsel et al., 2025) has been proposed as an alternative iterative scheme. By selecting iteration-dependent coefficients rather than a single fixed set, Polar Express significantly accelerates convergence during the early stages of the iteration.

**Polar Express.** Polar Express adopts the same iterative form as in Equation (6), but the coefficients are updated at each iteration rather than kept fixed. In particular, Polar Express determines the coefficients $a$, $b$, and $c$ by greedily solving a minimax optimization problem adapted to the evolving range of singular values at each step. The resulting iteration-dependent coefficients, which are pre-computed for efficiency, are summarized in Table 3 in Appendix D.

**Why do we prefer Polar Express?** Although the Jordan method has been successfully employed in the MUON optimizer, it may be less suitable as a general-purpose component for PAPO. Empirically, the Jordan iteration does not yield the exact polar factor $\mathbf{U}\mathbf{V}^\top$, but instead produces an approximation of the form $\mathbf{U}\mathbf{S}\mathbf{V}^\top$, where $\mathbf{S}$ is a diagonal matrix whose entries satisfy $\mathbf{S}_{ii} \sim \mathrm{Uniform}(0.5, 1.5)$. However, in PAPO this nonuniform scaling may undesirably alter the relative magnitudes of singular directions, potentially weakening the dominance of the mitigating-forgetting directions after applying Equation (4). By contrast, Polar Express attains a closer approximation to $\mathrm{polar}(\mathbf{G}_\tau^t)$ within a small number of iterations, producing a correction that is

closer to the identity on the dominant subspace. Because PAPO is intended as a broadly applicable plug-in for diverse continual learning methods, we adopt Polar Express as a safer and more stable choice in practice.

### 4.3. Plasticity Activation via Polar Operator

As discussed in Sections 4.1 and 4.2, we propose a method termed Plasticity Activation via Polar Operator (PAPO), which applies the polar operator to the gradient in order to enhance both stability and plasticity, as shown in Equation (4). To compute the polar factor in practice, we introduce the Polar Express approximation scheme, which replaces the use of SVD with a purely iterative procedure that relies only on matrix multiplications and additions.

To reduce computational overhead, we apply PAPO to the gradients periodically with an interval of $s$ epochs. Specifically, the gradients are rectified using Equation (4) before the parameter update only when the current epoch index is a multiple of $s$; otherwise, the original gradients are used without modification. To simplify the notation, $\mathrm{polar}(\cdot)$ in our algorithm denotes the iterative approximation defined by Equation (6), with coefficients selected according to Table 3 in Appendix D. The complete training procedure is summarized in Algorithm 1.

---

**Algorithm 1** Plasticity Activation via Polar Operator

**Require:** Task sequence $\mathcal{T} = \{T_1, T_2, \ldots, T_N\}$, balance coefficient $\lambda_1, \ldots \lambda_N$, polar interval $s$, training epochs $E$.
1: **Initialize:** Model parameters $\theta \leftarrow \theta_0$
2: **for** Task ID $\tau = 1$ **to** $N$ **do**
3:     **for** epoch $t = 1$ **to** $E$ **do**
4:         $\mathbf{G}_\tau^t \leftarrow \nabla_\theta \mathcal{L}(\theta_\tau^{t-1})$
5:         **if** $t \bmod s = 0$ **then**
6:             $\mathbf{G}_\tau^t \leftarrow \mathbf{G}_\tau^t + \lambda_\tau \, \mathrm{polar}(\mathbf{G}_\tau^t)$
7:         **end if**
8:         $\theta_\tau^t \leftarrow \theta_\tau^{t-1} - \eta\mathbf{G}_\tau^t$
9:     **end for**
10: **end for**
11: **Return** $\theta$

---

Next, we summarize the main advantages of our algorithm:

- PAPO is a plug-in module that can be seamlessly integrated into existing CL algorithms without modifying their loss functions or model architectures.

- PAPO relies exclusively on matrix multiplications and additions, making it well suited for efficient implementation on modern GPU hardware.

## 5. Experiments

In this section, we conduct extensive experiments to demonstrate the effectiveness of PAPO. The implementation details and additional experiments are provided Appendix E.

*Table 1.* ACC and BWT over all tasks on different datasets. Higher ACC and BWT indicate better generalization and less forgetting. All results are reproduced by us and averaged over 5 runs.

| Method | P-MNIST (10 tasks) | | CIFAR-100 (10 tasks) | | 5-Datasets (5 tasks) | | MiniImageNet (20 tasks) | |
|---|---|---|---|---|---|---|---|---|
| | ACC (%)↑ | BWT↑ | ACC (%)↑ | BWT↑ | ACC (%)↑ | BWT↑ | ACC (%)↑ | BWT↑ |
| ER | $87.24 \pm 0.53$ | $-0.11 \pm 0.01$ | $71.73 \pm 0.63$ | $-0.06 \pm 0.01$ | $88.31 \pm 0.22$ | $-0.04 \pm 0.00$ | $63.40 \pm 2.94$ | $-0.06 \pm 0.01$ |
| +PAPO | $88.18 \pm 0.42$ | $-0.10 \pm 0.00$ | $74.07 \pm 0.48$ | $-0.05 \pm 0.01$ | $90.30 \pm 0.33$ | $-0.04 \pm 0.00$ | $72.41 \pm 1.17$ | $-0.03 \pm 0.01$ |
| MAS | $86.98 \pm 0.84$ | $-0.04 \pm 0.01$ | $67.96 \pm 0.44$ | $-0.05 \pm 0.00$ | $88.73 \pm 0.79$ | $-0.04 \pm 0.01$ | $57.31 \pm 2.64$ | $-0.06 \pm 0.02$ |
| +PAPO | $88.15 \pm 0.26$ | $-0.03 \pm 0.01$ | $72.55 \pm 0.47$ | $-0.05 \pm 0.00$ | $90.72 \pm 0.18$ | $-0.00 \pm 0.00$ | $62.07 \pm 1.71$ | $-0.02 \pm 0.01$ |
| GPM | $93.18 \pm 0.14$ | $-0.03 \pm 0.00$ | $72.04 \pm 0.18$ | $-0.00 \pm 0.00$ | $91.12 \pm 0.20$ | $-0.01 \pm 0.00$ | $60.20 \pm 1.98$ | $-0.04 \pm 0.02$ |
| +PAPO | $94.00 \pm 0.27$ | $-0.03 \pm 0.00$ | $72.62 \pm 0.43$ | $+0.00 \pm 0.00$ | $92.00 \pm 0.07$ | $-0.01 \pm 0.00$ | $69.10 \pm 1.48$ | $+0.00 \pm 0.00$ |
| TRGP | $96.34 \pm 0.11$ | $-0.01 \pm 0.00$ | $74.46 \pm 0.32$ | $-0.01 \pm 0.00$ | $93.56 \pm 0.10$ | $-0.00 \pm 0.00$ | $61.78 \pm 0.60$ | $-0.01 \pm 0.01$ |
| +PAPO | $96.73 \pm 0.10$ | $-0.01 \pm 0.00$ | $75.65 \pm 0.17$ | $+0.00 \pm 0.00$ | $94.04 \pm 0.13$ | $-0.00 \pm 0.00$ | $70.97 \pm 0.82$ | $-0.00 \pm 0.00$ |

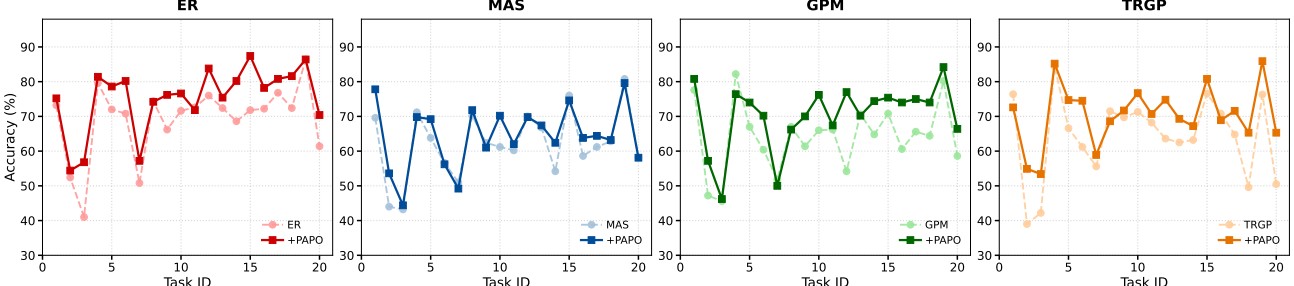

*Figure 3.* Performance comparison on the MiniImageNet dataset.

### 5.1. Experimental Setup

**Datasets and network architectures.** For *vision* tasks we follow the dataset and architecture configurations of Saha et al. (2021). We evaluate on four standard CL benchmarks: Permuted MNIST (PMNIST) (Lecun et al., 2002), 10-split CIFAR-100 (Krizhevsky, 2009), 20-split MiniImageNet (Vinyals et al., 2016), and the 5-Datasets sequence (Ebrahimi et al., 2020). For PMNIST we use a fully connected network with two hidden layers of 100 units each. For Split CIFAR-100 we adopt a 5-layer AlexNet. For Split MiniImageNet and the 5-Datasets sequence we use a reduced ResNet-18. For *language* tasks, we follow the setup of Liang et al. (2026) and evaluate on the SuperNI benchmark (Wang et al., 2022b), which comprises 15 tasks and two task orders (Order 1 and Order 2). For SuperNI benchmark we use Llama-2-7B (Touvron et al., 2023) as the backbone model. Further dataset and implementation details are provided in Appendix E.1.

**Baselines.** For vision tasks, we evaluate PAPO when applied to representative methods from three categories of CL: a replay-based method (ER with reservoir sampling (Chaudhry et al., 2019b)), a regularization-based method (MAS (Aljundi et al., 2018)), and orthogonal-projection-based methods (GPM (Saha et al., 2021) and TRGP (Lin et al., 2022)). For language experiments, we apply PAPO to InfLoRA (Liang & Li, 2024) and GainLoRA (Liang et al., 2026). Detailed hyperparameter settings are reported in

Appendix E.2.

**Evaluation metrics.** For a fair comparison, we adopt the same evaluation metrics as in Saha et al. (2021) and Liang et al. (2026). We report three standard metrics: average accuracy (ACC), backward transfer (BWT) and forgetting (FT). For vision tasks we report ACC and BWT; for language tasks we report ACC and FT. Let $A_{t,i}$ denote the test accuracy on task $i$ after training up to task $t$, and let $T$ be the total number of tasks. The metrics are defined as: $\text{ACC} = \frac{1}{T} \sum_{i=1}^{T} A_{T,i}, \text{BWT} = \frac{1}{T-1} \sum_{i=1}^{T-1} (A_{T,i} - A_{i,i}), \text{FT} = \frac{1}{T-1} \sum_{i=1}^{T-1} (\max_{t \in \{1,...,T-1\}} A_{t,i} - A_{T,i})$.

**Other Experiments.** We also conduct experiments in the non-CL setting in Appendix F.5, in the online-CL setting in Appendix F.6, and include comparisons with additional methods in Appendix F.7.

### 5.2. Performance Comparison

#### 5.2.1. RESULTS ON VISION TASKS

**ACC and BWT.** We first investigate how incorporating PAPO into CL methods affects ACC and BWT. As shown in Table 1, PAPO yields consistent improvements across all baselines. In particular, averaged over the four benchmarks, incorporating PAPO into ER, MAS, GPM, and TRGP results in accuracy gains of $3.52\%$, $3.18\%$, $2.80\%$, and $2.91\%$, with corresponding BWT increases of $0.01$, $0.02$, $0.01$, and $0.01$, respectively. The effectiveness of PAPO is espe-

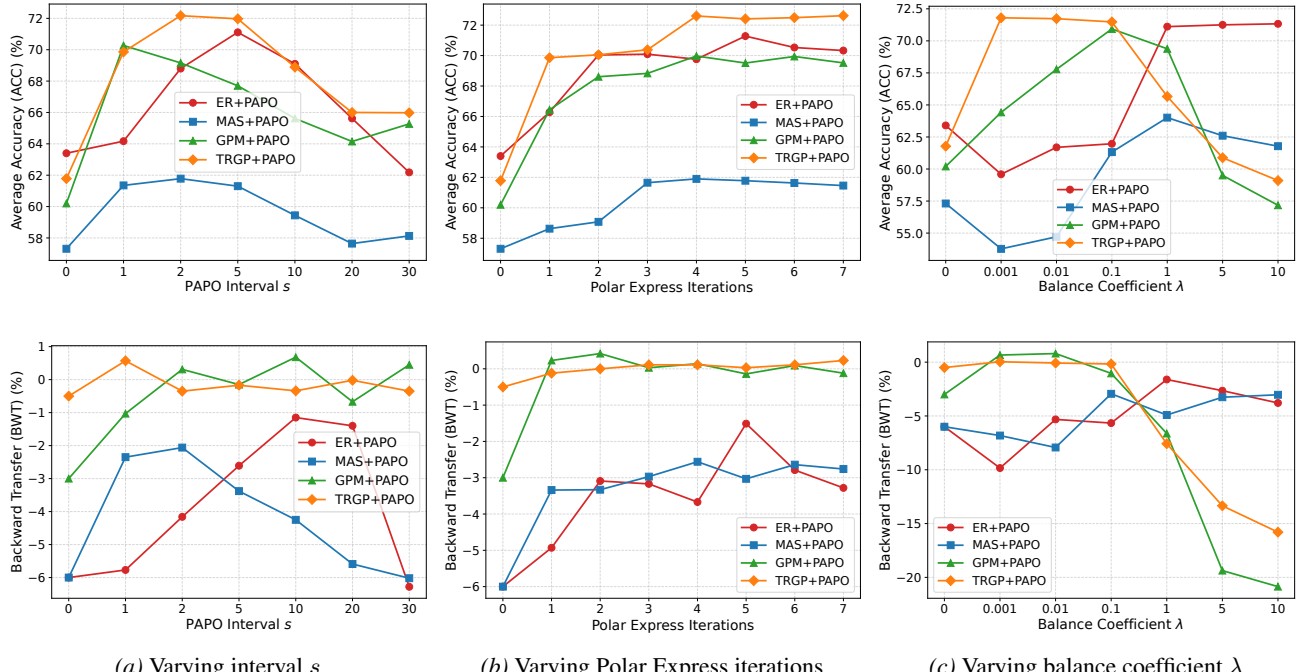

*(a) Varying interval $s$*    *(b) Varying Polar Express iterations*    *(c) Varying balance coefficient $\lambda$*

*Figure 4.* **Ablation study of PAPO hyperparameters on the MiniImageNet dataset.** The top row reports the ACC, while the bottom row reports the BWT. The results illustrate the effect of **(a)** the PAPO application interval $s$, **(b)** the number of Polar Express iterations, and **(c)** the balance coefficient $\lambda$ on model performance.

cially pronounced on the MiniImageNet benchmark, where it improves the accuracy of ER, MAS, GPM, and TRGP by 9.01%, 4.76%, 8.90%, and 9.19%, respectively, while simultaneously alleviating forgetting with BWT improvements of 0.03, 0.04, 0.04, and 0.01. We note that these datasets are widely studied benchmarks in CL. Therefore, achieving consistent improvements on them highlights the practical significance of PAPO.

**Plasticity.** To assess the plasticity of CL methods, we examine the test accuracy on each task after training on successive tasks on the MiniImageNet benchmark. As shown in Figure 3, across all four subfigures the curves with darker colors and square markers generally lie above the corresponding curves with lighter colors and circular markers. This observation suggests that incorporating PAPO into ER, MAS, GPM, and TRGP tends to improve test accuracy on individual tasks, thereby enhancing model plasticity.

We also compare the gradient singular value spectra produced by ER, MAS and GPM with and without PAPO in Appendix F.3, highlighting that incorporating PAPO alleviates singular value collapse.

### 5.2.2. RESULTS ON LANGUAGE TASKS

As shown in Table 2, averaged over Order 1 and Order 2, incorporating PAPO into InfLoRA and GainLoRA yields accuracy gains of 3.37% and 1.24%, respectively, with cor-

*Table 2.* ACC and FT on the SuperNI benchmark (Order 1 and Order 2) using the Llama-2-7B model. Higher ACC and lower FT indicate better generalization and less forgetting.

| Methods | Order 1 | | Order 2 | |
|---|---|---|---|---|
| | ACC↑ | FT↓ | ACC↑ | FT↓ |
| InfLoRA | 42.93 | 11.23 | 39.94 | 15.00 |
| +PAPO | 48.08 | 6.29 | 41.53 | 13.51 |
| GainLoRA | 51.27 | 2.84 | 50.17 | 4.71 |
| +PAPO | 52.47 | 2.09 | 51.45 | 2.69 |

responding reductions in forgetting of 3.22% and 1.39%. We further analyze the computational cost introduced by PAPO in Appendix F.4, highlighting that PAPO is computationally efficient.

### 5.3. Ablation Study

**Interval $s$.** Figure 4a illustrates the effect of periodically applying PAPO with interval $s$ (in epochs) on ACC and BWT. Applying PAPO at every epoch, that is $s = 1$, increases computational cost and does not consistently yield the best performance. In contrast, a large interval $s$ fails to activate the previously suppressed directions sufficiently often; in this regime, PAPO behaves more like a sporadic perturbation that abruptly alters the update direction. As a result, performance degrades noticeably when $s \geq 10$. In particular, when $s = 30$, applying PAPO to ER even

leads to worse performance than the baseline without PAPO. Empirically on miniImageNet, an intermediate interval $s$ in the range of 1–5 provides a favorable trade-off between computational overhead and improvements in both ACC and BWT across the evaluated methods.

**Polar Express iterations.** Figure 4b illustrates the effect of the number of Polar Express iterations used to approximate $\mathrm{polar}(\mathbf{G}_\tau^t)$ on ACC and BWT. Both metrics improve rapidly in the first few iterations and then plateau. Notably, even a single iteration yields a clear performance gain. In practice, 2–4 iterations are sufficient to capture most of the benefits while maintaining computational efficiency. This trend aligns with the expectation that a small number of iteration-dependent updates can substantially enhance performance compared with a zero-iteration baseline.

**Balance coefficient $\lambda$.** The balance coefficient $\lambda$ directly affects the singular values of the gradient after applying PAPO, as each singular value $\sigma_i$ is shifted to $\sigma_i + \lambda$. Consequently, $\lambda$ controls the relative strength of the dominant mitigating-forgetting directions in the updated gradient. If $\lambda$ is excessively large, it can dominate the original spectrum and thereby disturb these dominant directions. This effect is more pronounced when the original singular values are small, as is often the case for orthogonal-projection-based methods, since in this regime $\sigma_i + \lambda \approx \lambda$ for all $i$. As shown in Figure 4c, when $\lambda < 1$, both ACC and BWT for TRGP and GPM improve; however, when $\lambda \geq 1$, their performance degrades, and at $\lambda = 10$ these methods exhibit severe forgetting. This observation corroborates Remark 4.2. If the dominant role of the mitigating-forgetting directions is disrupted, the model's stability is compromised and forgetting may increase.

## 6. Conclusion

In this work, we first revisit the singular value spectra of gradients in three representative CL algorithms that are designed to mitigate forgetting, and we show that these methods consistently exhibit singular value collapse. We then propose Plasticity Activation via the Polar Operator (PAPO), a mechanism that preserves the forgetting-mitigation properties of the base methods while improves plasticity. Finally, we evaluate PAPO on both vision and language benchmarks and demonstrate that incorporating PAPO into existing CL methods yields higher accuracy and reduced forgetting.

## Acknowledgment

This work is supported by National Key R&D Projects (NO. 2024YFC3307100), NSFC Grant (No. 62576364), GuangDong Basic and Applied Basic Research Foundation (2026B1515020071), Shenzhen Basic Research Project (Natural Science Foundation) Basic Research Key Project (NO. JCYJ20241202124430041), and Shenzhen Science and Technology Program (NO.SYSRD202505529113401002). The computation is completed in the HPC Platform of Huazhong University of Science and Technology.

## Impact Statement

This paper presents work whose goal is to advance the field of machine learning. There are many potential societal consequences of our work, none of which we feel must be specifically highlighted here.

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

## A. Related work

**Mitigate forgetting.** CL has achieved substantial empirical progress, with existing approaches broadly categorized into three families: (1) *Regularization-based methods*, which introduce explicit penalty terms to restrict updates on parameters important for previous tasks (Kirkpatrick et al., 2017; Zenke et al., 2017), or employ knowledge distillation by aligning the predictions of the current model (student) with those of the previous model (teacher) to mitigate forgetting (Li & Hoiem, 2017; Dhar et al., 2019; Fostiropoulos et al., 2023); (2) *Replay-based methods*, which either directly store data or use a diffusion model (Lu et al., 2025b; Zhang et al., 2025) to replay data from past tasks during training on new tasks (Chaudhry et al., 2019a; Riemer et al., 2018; Buzzega et al., 2020; Yang et al., 2023b; Gao & Liu, 2023; Kim et al., 2024; Wang et al., 2025), or retain gradient or feature information from previous tasks and enforce new updates to remain orthogonal to past gradients, thereby avoiding explicit data replay (Farajtabar et al., 2020; Saha et al., 2021; Lin et al., 2022; Yang et al., 2023a; 2025). (3) *Architecture-based methods* (Rusu et al., 2016; Yoon et al., 2018; Wang et al., 2022c), which dynamically expand or adapt the network architecture to preserve knowledge from earlier tasks. Recently, gradient modification-based methods (Bian et al., 2024; Li et al., 2026; Feng et al., 2025) have also played an important role in improving model performance in CL.

**Stability–plasticity trade-off.** The trade-off between stability and plasticity was famously framed by Carpenter & Grossberg (1987) as the *stability–plasticity dilemma*: a fundamental tension between retaining performance on past tasks and adapting to new ones. Recent work has proposed diverse strategies to address this dilemma in CL. We categorize these strategies into three classes: architecture-based methods, methods tailored to specific CL algorithms, and methods tailored to specific CL paradigms. (1) Architecture-based approaches decouple components or add auxiliary modules to reconcile stability and plasticity. For example, Wang et al. (2022a) propose an energy-based framework that mitigates conflicts in dynamic architectures by decoupling the training of independent modules while ensuring bidirectional compatibility. Kim et al. (2023) introduce Auxiliary Network Continual Learning (ANCL), which employs a dual-network design in which a primary network preserves stability and an auxiliary network promotes plasticity via a learnable regularizer. Lu et al. (2025a) show that depth tends to favor plasticity while width favors stability and they propose the DualArch framework to separate these objectives using two specialized lightweight networks. (2) Methods that target specific CL algorithms include Kong et al. (2022)'s Advanced Null Space (AdNS), which uses low-rank approximation to derive a null space for gradient projection, and Yang et al. (2026)'s data-free dual orthogonal projection scheme for continual model merging. (3) Approaches designed for particular CL paradigms include Chen et al. (2023), who develop an excess-risk bound together with a dynamic optimization algorithm for continual meta-learning, and Elsayed & Mahmood (2024) propose Utility-based Perturbed Gradient Descent (UPGD), a method for streaming CL that protects useful units with small updates while rejuvenating less useful ones via larger perturbations. These prior solutions often add architectural complexity or remain specialized to particular algorithms or scenarios. As a result they do not offer a broadly applicable remedy to the stability and plasticity dilemma. In contrast, PAPO is an architecture-free plug-in that can be incorporated seamlessly into a wide range of CL algorithms.

## B. Limitations

In this work we propose PAPO, a general method to enhance plasticity in CL algorithms that mitigate forgetting. PAPO depends on a coefficient $\lambda$ that must be set manually. Developing an adaptive scheme for $\lambda$ would increase PAPO's performance. In future work we plan to apply PAPO to multimodal large language models (MLLMs) and to integrate it with additional CL methods.

## C. Proof of Theorem 3.1

*Proof.* First, we prove the non-strict inequality. Recall the rank inequality for matrix multiplication: for any matrices $A$ and $B$, $\text{rank}(AB) \leq \min\{\text{rank}(A), \text{rank}(B)\}$. By setting $A = P_{\mathcal{X}_m}^\perp$ and $B = G$, we immediately have

$$\text{rank}(P_{\mathcal{X}_m}^\perp G) \leq \text{rank}(G). \tag{7}$$

Next, we prove the strict one. Assume the intersection of the column space of $G$ (denoted as $\text{Col}(G)$) and $\mathcal{X}_m$ is non-trivial. This implies that there exists a non-zero vector $v \in \text{Col}(G) \cap \mathcal{X}_m$.

Since $v \in \text{Col}(G)$, there exists a coefficient vector $w \in \mathbb{R}^d$ such that $v = Gw$. Note that $w \notin \ker(G)$ because $Gw = v \neq 0$.

Since $v \in \mathcal{X}_m$ and $P_{\mathcal{X}_m}^{\perp}$ projects onto the orthogonal complement of $\mathcal{X}_m$, we have $P_{\mathcal{X}_m}^{\perp} v = 0$.

Combining these, we have

$$(P_{\mathcal{X}_m}^{\perp} G)w = P_{\mathcal{X}_m}^{\perp}(Gw) = P_{\mathcal{X}_m}^{\perp} v = 0. \tag{8}$$

This shows that $w \in \ker(P_{\mathcal{X}_m}^{\perp} G)$. Consequently, the kernel of $G$ is a proper subspace of the kernel of $P_{\mathcal{X}_m}^{\perp} G$ (i.e., $\ker(G) \subsetneq \ker(P_{\mathcal{X}_m}^{\perp} G)$).

By the Rank-Nullity Theorem, $\mathrm{rank}(A) = d - \dim(\ker(A))$. Since $\dim(\ker(P_{\mathcal{X}_m}^{\perp} G)) > \dim(\ker(G))$, it follows that

$$\mathrm{rank}(P_{\mathcal{X}_m}^{\perp} G) < \mathrm{rank}(G). \tag{9}$$

The proof is complete. $\qquad\square$

## D. Polar operator

*Table 3.* Coefficient settings for Polar Express. The reported values are taken from Amsel et al. (2025).

| $k$ | $a$ | $b$ | $c$ |
|---|---|---|---|
| 1 | 8.28721 | $-23.5959$ | 17.3004 |
| 2 | 4.10706 | $-2.94785$ | 0.544843 |
| 3 | 3.94869 | $-2.9089$ | 0.551819 |
| 4 | 3.31842 | $-2.48849$ | 0.510049 |
| 5 | 2.30065 | $-1.6689$ | 0.418807 |
| 6 | 1.8913 | $-1.268$ | 0.376804 |
| 7 | 1.875 | $-1.25$ | 0.375 |
| 8 | 1.875 | $-1.25$ | 0.375 |

## E. Experimental Details and Additional Experiments

### E.1. Dataset and Network Architecture

We summarize the datasets and network architectures used in our experiments. Details for vision tasks are provided in Section E.1.1, and details for language tasks are provided in Section E.1.2.

#### E.1.1. VISION TASK DATASETS AND NETWORK ARCHITECTURES

We follow the dataset and network architecture configurations of Saha et al. (2021).

**Dataset.** The PMNIST dataset consists of 10 sequential tasks generated by applying different random permutations to the pixels of the original MNIST images. The 10-Split CIFAR-100 dataset is constructed by partitioning the 100 classes of CIFAR-100 into 10 disjoint tasks with 10 classes per task. The 20-Split miniImageNet dataset is constructed by splitting the 100 classes of miniImageNet into 20 sequential tasks, each containing 5 classes. Finally, we evaluate on a sequence of five datasets, including CIFAR-10, MNIST, SVHN, notMNIST, and Fashion-MNIST, where classification on each dataset is treated as a separate task. No data augmentation is applied in any experiment.

**Network Architecture.** AlexNet-like architecture: This is the same architecture used by (Serra et al., 2018) with batch normalization added in each layer except the classifier layer. The network consists of 3 convolutional layers of 64, 128, and 256 filters with $4 \times 4$, $3 \times 3$, and $2 \times 2$ kernel sizes, respectively, plus two fully connected layers of 2048 units each. Rectified linear units is used as activations, and $2 \times 2$ max-pooling after the convolutional layers. Dropout of 0.2 is used for the first two layers and 0.5 for the rest. Reduced ResNet18 architecture: This is the similar architecture used by (Lopez-Paz & Ranzato, 2017). For miniImageNet experiment, we use convolution with stride 2 in the first layer. For both miniImageNet and 5-Datasets experiments we replace the $4 \times 4$ average-pooling before classifier layer with $2 \times 2$ average-pooling. All the networks use ReLU in the hidden units and softmax with cross entropy loss in the final layer.

#### E.1.2. LANGUAGE TASK DATASETS

We follow the dataset and network architecture configurations of Liang et al. (2026).

**Dataset.** We use the SuperNI benchmark as our language CL benchmark. SuperNI comprises a diverse set of language tasks, including dialogue generation, information extraction, question answering, summarization, and sentiment analysis. Following the protocol of Zhao et al. (2024), we select three tasks from each category, resulting in a total of 15 tasks. These tasks are organized into two different task sequences with distinct orders, referred to as Order 1 and Order 2.

### E.2. Hyperparameter settings

In this section we list the hyperparameter settings for vision (Table 4) and language (Table 5) experiments. For a fair comparison, we adopt the original hyperparameter configurations used by Saha et al. (2021) and Lin et al. (2022) for the vision benchmarks and by Liang et al. (2026) for the language benchmarks.

Following Liang et al. (2026), language tasks are implemented with instruction tuning and optimized using AdamW (Loshchilov & Hutter, 2019). We incorporate LoRA adapters into the query and value projections of the multi-head attention in each Transformer block. The global batch size is set to 32 and each task is trained for 50 epochs. For GainLoRA, we follow Liang et al. (2026) and use InfLoRA to update the newly introduced LoRA branch.

*Table 4.* Hyperparameter settings for vision tasks. $\lambda_0$ denotes the PAPO coefficient for the first task, $\lambda$ denotes the PAPO coefficient for tasks starting from the second one. The scale factor $\gamma$ controls a geometric decay of the PAPO coefficient: for task $t \geq 2$ the coefficient is $\lambda / \gamma^{t-2}$. Experiments were run with seeds 1–5 and the reported results are averaged.

| Setting | Hyperparameter | P-MNIST | CIFAR-100 | 5-Datasets | MiniImageNet |
|---------|----------------|---------|-----------|------------|--------------|
| **Shared** | Task nums | 10 | 10 | 5 | 20 |
| | Network | MLP | AlexNet | ResNet-18 | ResNet-18 |
| | Epochs | 5 | 200 | 100 | 100 |
| | Learning rate | 0.01 | 0.05 | 0.1 | 0.1 |
| | Batch size | 10 | 64 | 64 | 64 |
| **ER** | Buffer size | 1000 | 2000 | 3000 | 500 |
| | Interval | 2 | 5 | 5 | 5 |
| | $\lambda_0$ | 0.01 | 0.5 | 1 | 5 |
| | $\lambda$ | 0.01 | 0.5 | 1 | 5 |
| | Scale factor $\gamma$ | 1 | 1 | 5 | 1 |
| **MAS** | MAS coefficient | 1 | 1 | 1 | 1 |
| | Interval | 5 | 5 | 2 | 2 |
| | $\lambda_0$ | 10 | 5 | 5 | 10 |
| | $\lambda$ | 0.001 | 5 | 5 | 10 |
| | Scale factor $\gamma$ | 1 | 1 | 1 | 1 |
| **GPM** | Interval | 2 | 2 | 2 | 2 |
| | $\lambda_0$ | 0.1 | 5 | 1 | 5 |
| | $\lambda$ | 0.0001 | 0.005 | 0.05 | 0.05 |
| | Scale factor $\gamma$ | 1 | 2 | 2 | 1 |
| **TRGP** | Interval | 3 | 2 | 2 | 5 |
| | $\lambda_0$ | 0.01 | 5 | 5 | 5 |
| | $\lambda$ | 0.0001 | 0.01 | 0.01 | 0.05 |
| | Scale factor $\gamma$ | 1 | 1 | 2 | 1 |

*Table 5.* Hyperparameter settings for language tasks. We report only the hyperparameters used by PAPO and keep all other settings identical to Liang et al. (2026).

| Setting | Hyperparameter | Order1 | Order2 |
|---|---|---|---|
| **InfLoRA** | Interval | 1 | 5 |
| | $\lambda$ | 0.5 | 0.5 |
| **GainLoRA** | Interval | 5 | 5 |
| | $\lambda$ | 0.005 | 0.005 |

# F. Additional Experiments

## F.1. Singular Value spectra of ER, MAS, and GPM

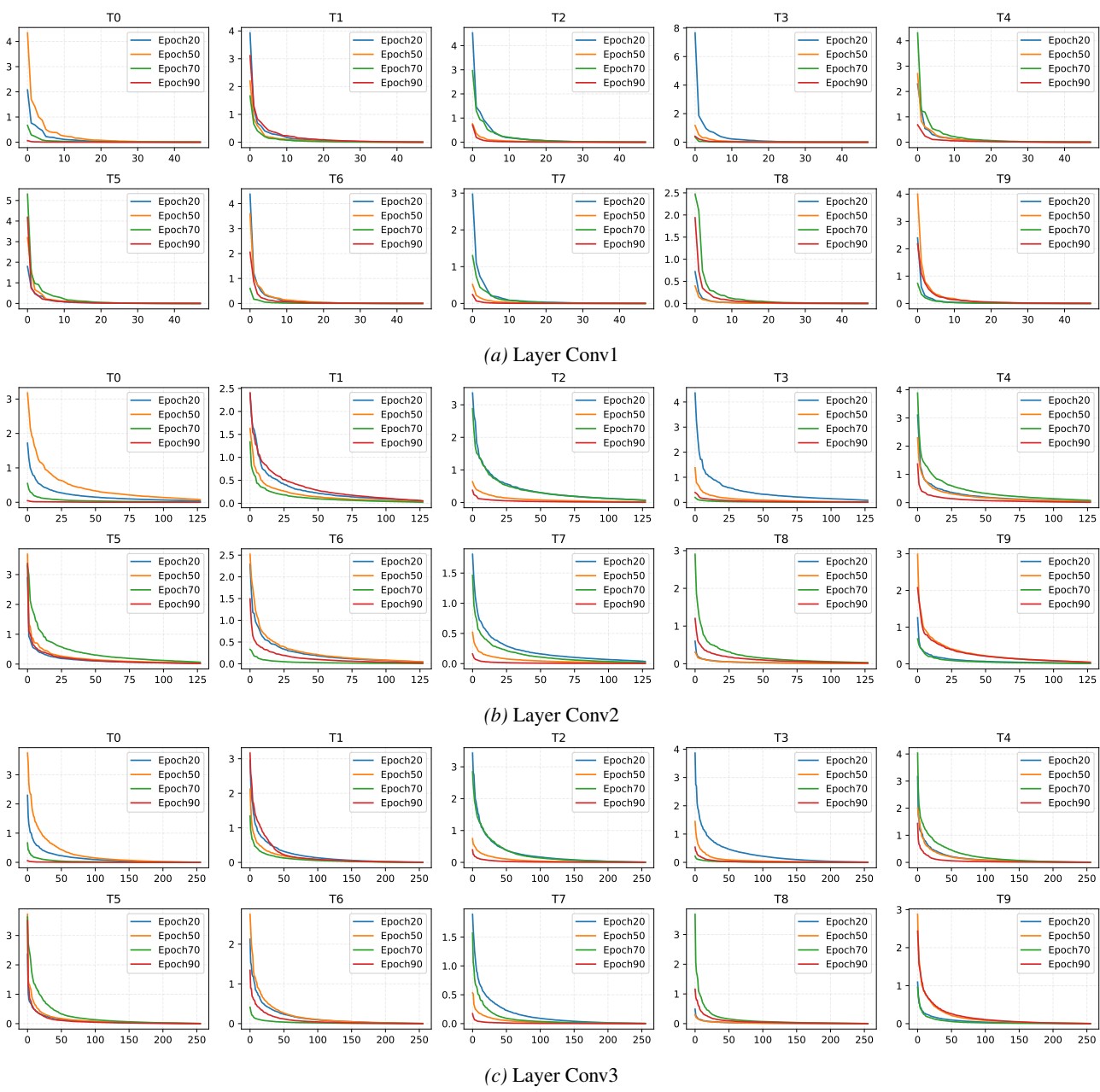

*Figure 5.* Singular value spectra for convolutional (Conv) layers Conv1, Conv2 and Conv3 at epochs 20, 50, 70 and 90 across tasks T0–T9 in **ER**. These results were obtained on the CIFAR-100 benchmark using an AlexNet backbone. Singular value collapse is also evident across multiple epochs.

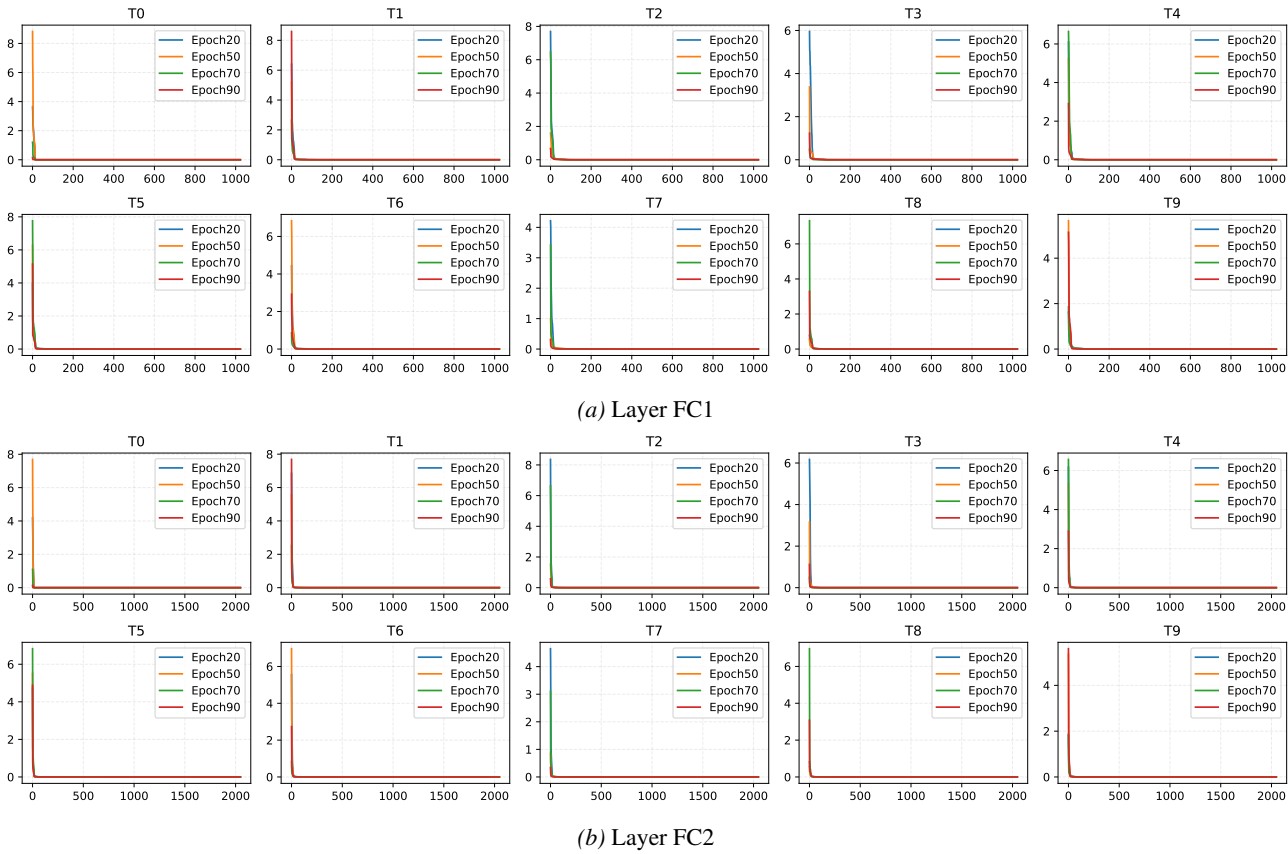

*(a)* Layer FC1

*(b)* Layer FC2

*Figure 6.* Singular value spectra for fully connected (FC) layers FC1 and FC2 at epochs 20, 50, 70 and 90 across tasks T0–T9 under **ER**. Results were obtained on the CIFAR-100 benchmark using an AlexNet backbone. The FC layers exhibit pronounced singular value collapse, characterized by an abrupt spectral drop and a long tail of near-zero singular values.

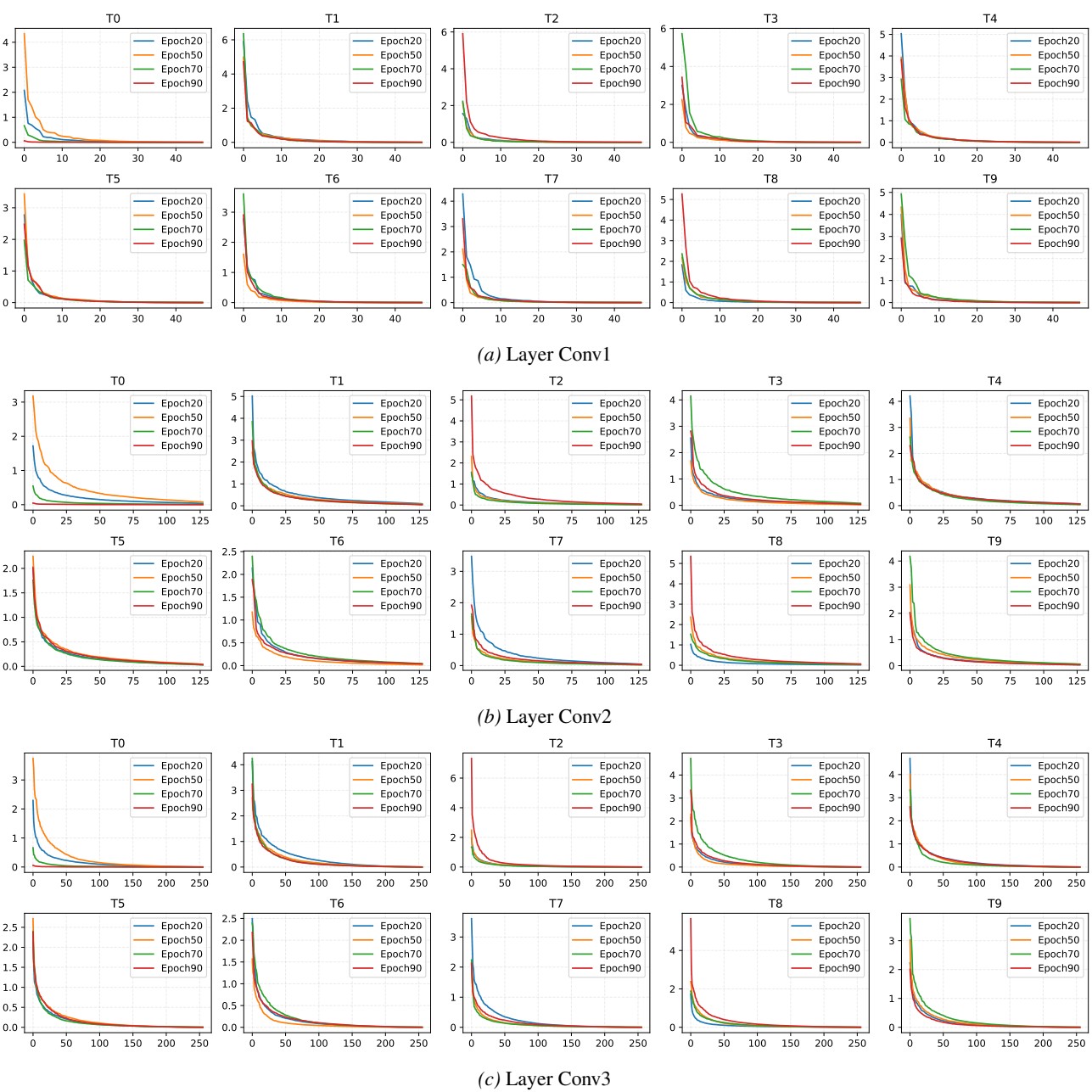

*(a)* Layer Conv1

*(b)* Layer Conv2

*(c)* Layer Conv3

*Figure 7.* Singular value spectra for convolutional layers Conv1, Conv2 and Conv3 at epochs 20, 50, 70 and 90 across all tasks T0 to T9 in **MAS**. These results were obtained on the CIFAR-100 benchmark using an AlexNet backbone. Singular value collapse is also evident across multiple epochs.

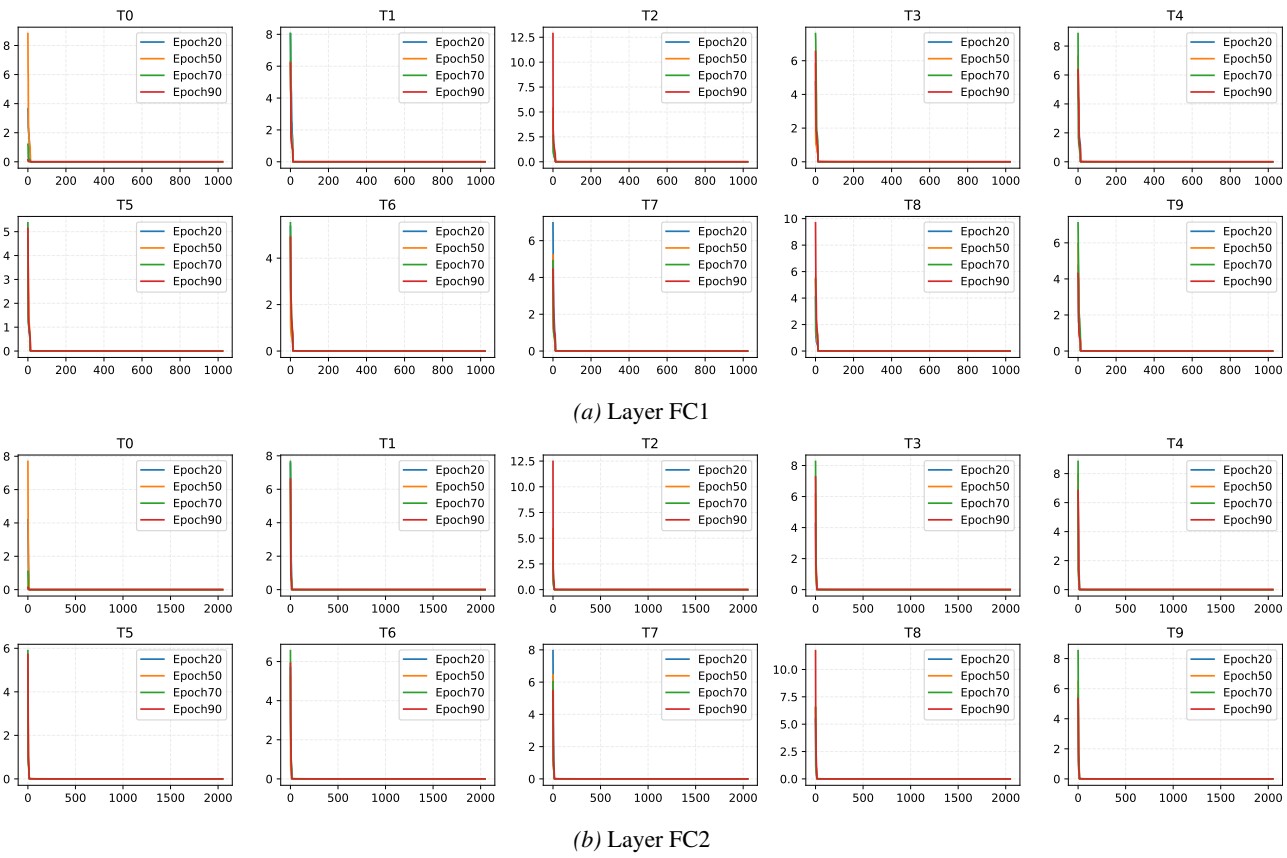

*Figure 8.* Singular value spectra for fully connected (FC) layers FC1 and FC2 at epochs 20, 50, 70 and 90 across tasks T0–T9 under **MAS**. Results were obtained on the CIFAR-100 benchmark using an AlexNet backbone. The FC layers exhibit pronounced singular value collapse, characterized by an abrupt spectral drop and a long tail of near-zero singular values.

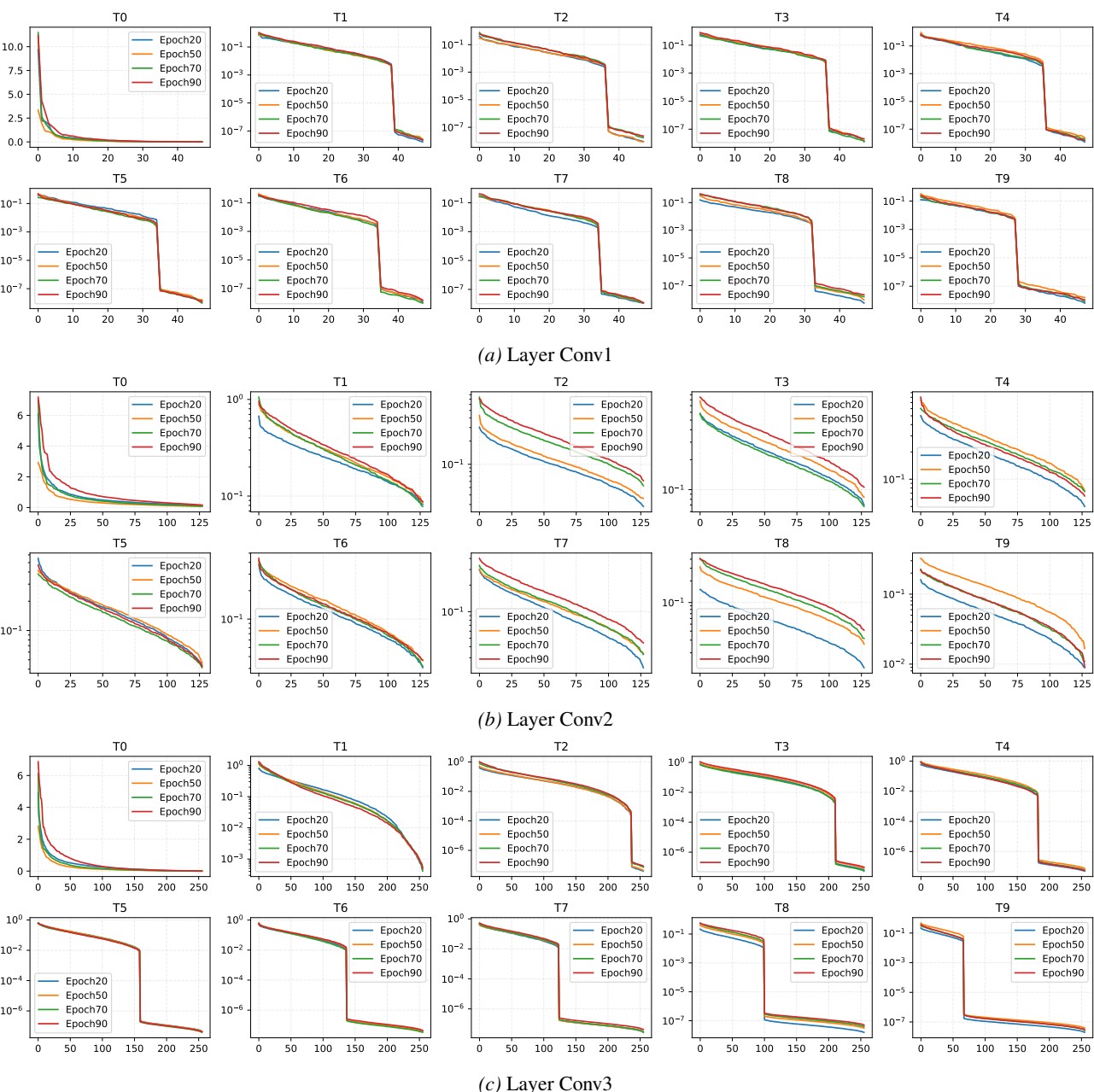

*(a)* Layer Conv1

*(b)* Layer Conv2

*(c)* Layer Conv3

*Figure 9.* Singular value spectra for convolutional layers Conv1, Conv2 and Conv3 at epochs 20, 50, 70 and 90 across tasks T0–T9 under **GPM**. Results were obtained on the CIFAR-100 benchmark using an AlexNet backbone. To reveal the spectral cliff, the vertical axis is plotted on a logarithmic scale for tasks T1–T9. We observe a pronounced spectral cliff in Conv1 and Conv3. In Theorem 3.1 we show that this pattern arises when the intersection between the column space of the gradient and the subspace spanned by vectors from previous tasks is non-trivial. By contrast, Conv2 does not exhibit similar overlap with previous tasks.

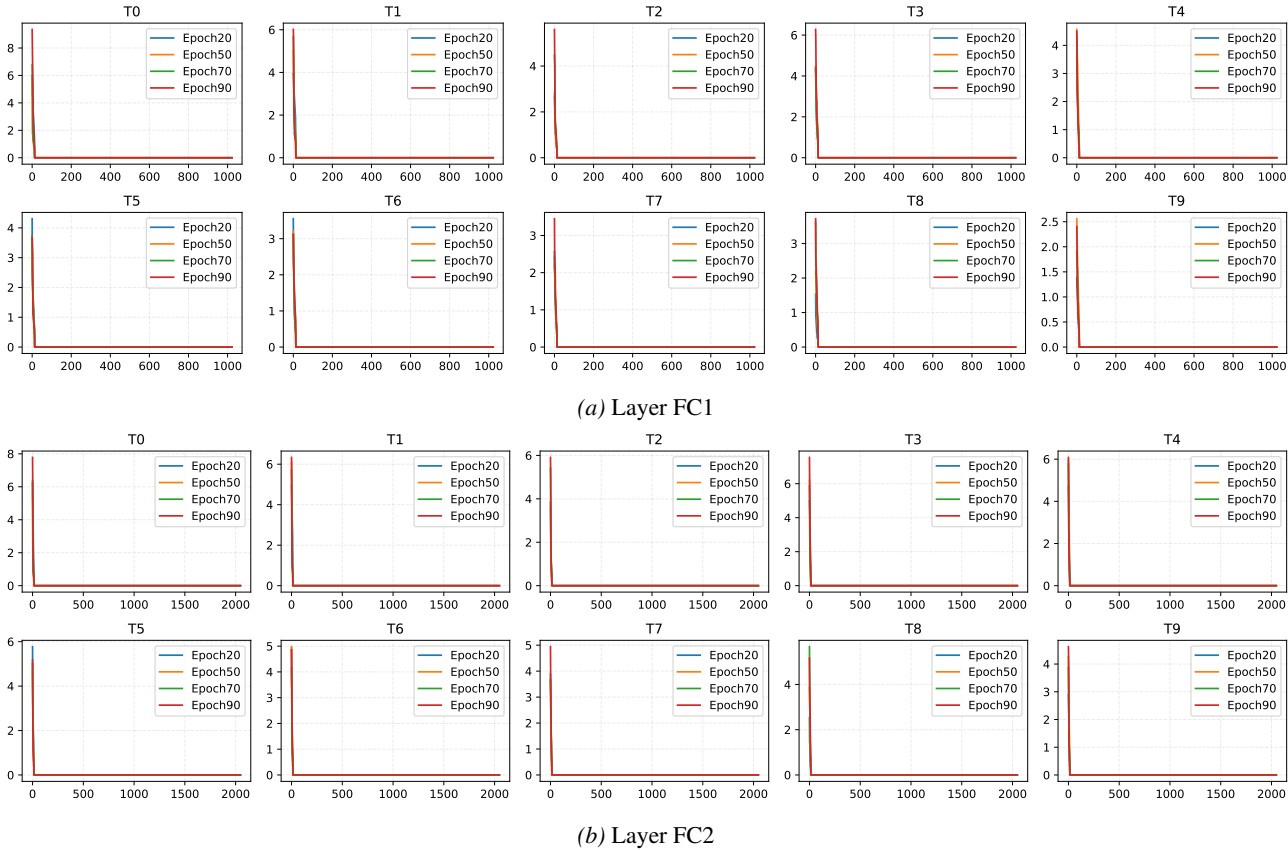

*(a)* Layer FC1

*(b)* Layer FC2

*Figure 10.* Singular value spectra for fully connected (FC) layers FC1 and FC2 at epochs 20, 50, 70 and 90 across tasks T0–T9 under **GPM**. Results were obtained on the CIFAR-100 benchmark using an AlexNet backbone. The FC layers exhibit pronounced singular value collapse, characterized by an abrupt spectral drop and a long tail of near-zero singular values.

## F.2. Singular value spectra for different GPM thresholds

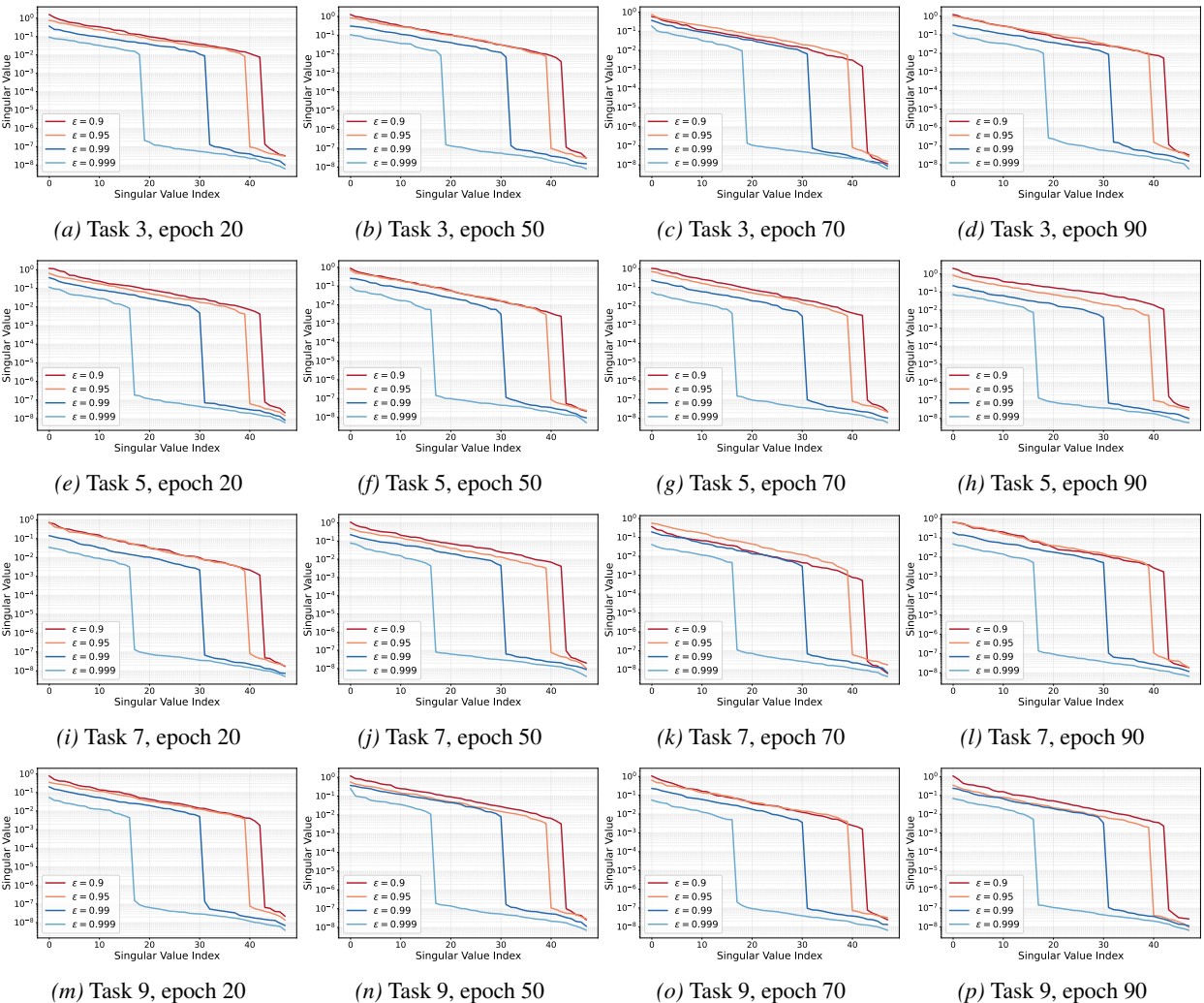

*Figure 11.* Conv1 singular value spectra under different GPM thresholds (i.e., 0.9, 0.95, 0.99 and 0.999). The horizontal axis denotes the singular value index, and the vertical axis shows the singular values on a logarithmic scale. For notational simplicity, we denote the projection threshold by $\epsilon$. At any epoch (i.e.,20, 50, 70 and 90) and for any task (i.e., 3, 5, 7, 9), the spectral cliff shifts leftward as $\epsilon$ increases, indicating that larger projection thresholds induce more severe singular value collapse.

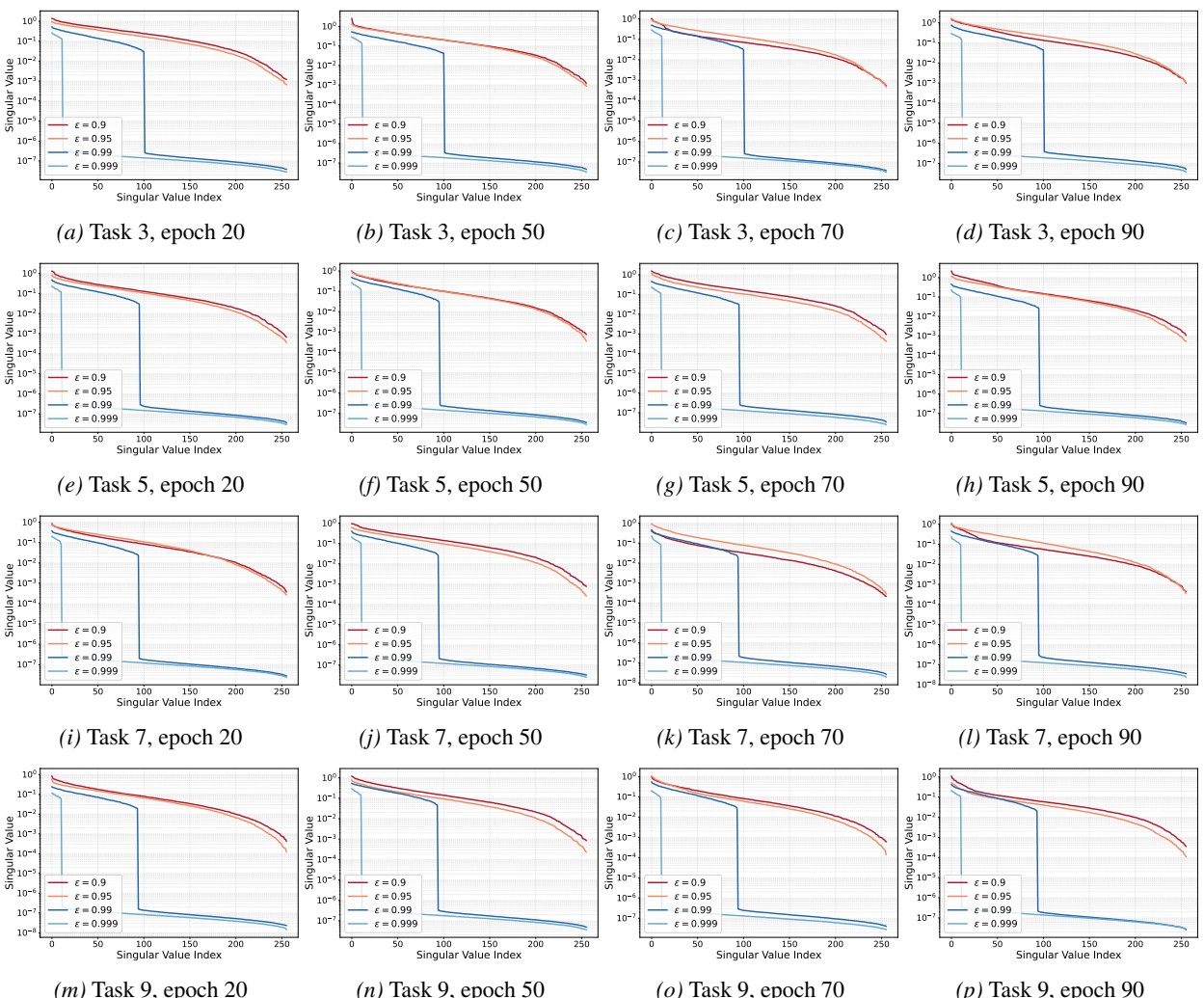

*Figure 12.* Conv3 singular value spectra under different GPM thresholds (i.e., 0.9, 0.95, 0.99 and 0.999). The horizontal axis denotes the singular value index, and the vertical axis shows the singular values on a logarithmic scale. At any epoch (i.e.,20, 50, 70 and 90) and for any task (i.e., 3, 5, 7, 9), the spectral cliff shifts leftward as $\epsilon$ increases, indicating that larger projection thresholds induce more severe singular value collapse.

### F.3. Comparison of Singular Value Spectra With and Without PAPO

In this section, we compare the gradient singular value spectra produced by ER, MAS and GPM when run without PAPO and when augmented with PAPO to assess PAPO's effect on singular-value collapse. We report results for epochs 20, 50, 70 and 90 for tasks T0, T3, T5, T7 and T9 in Figures 13 and 14. We observe that applying PAPO to these methods yields a clear recovery of singular values away from the near-zero regime. In particular, the bottom of the singular value cliff for GPM is lifted when PAPO is applied. Thus, incorporating PAPO mitigates singular value collapse.

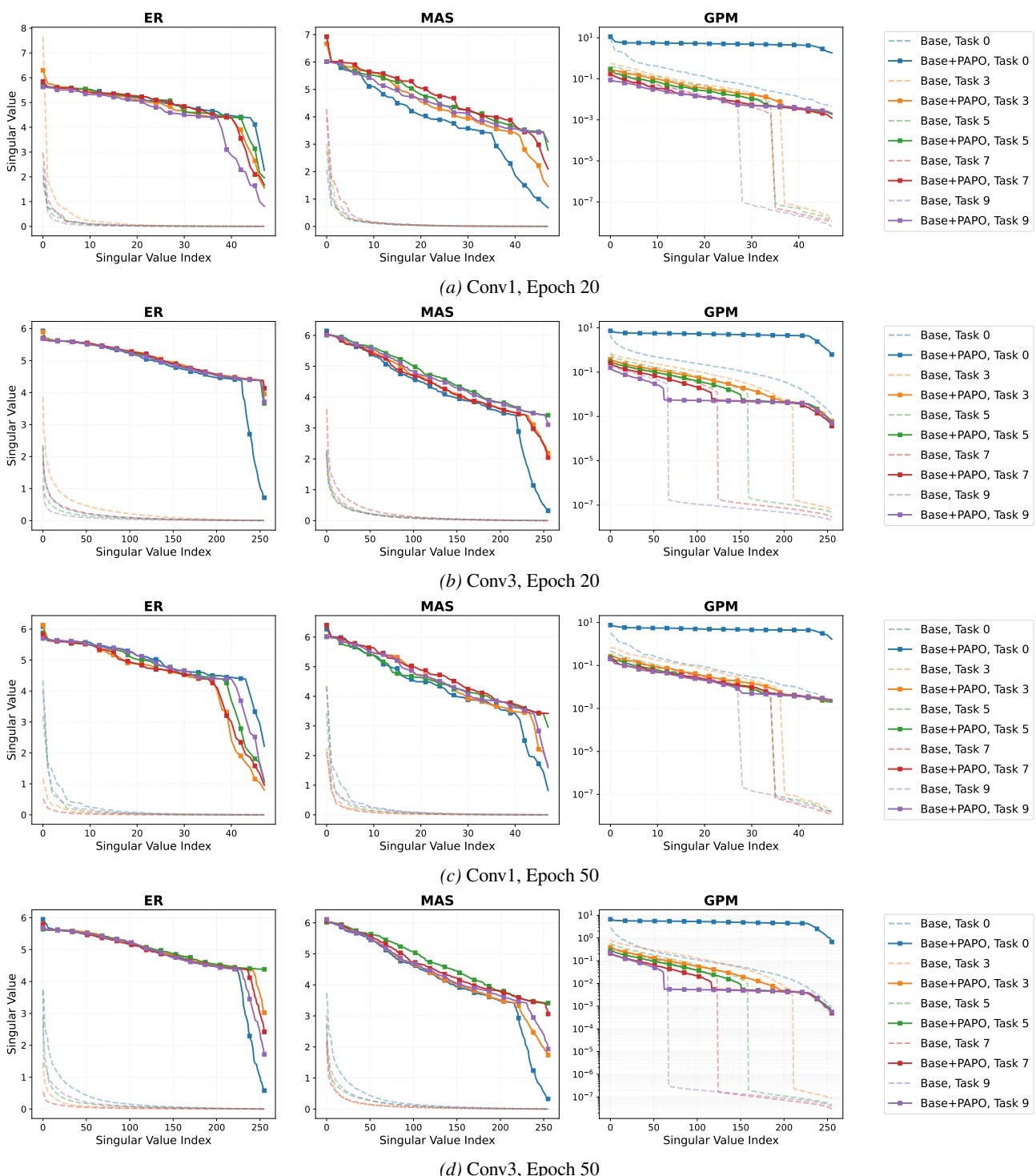

*(a)* Conv1, Epoch 20

*(b)* Conv3, Epoch 20

*(c)* Conv1, Epoch 50

*(d)* Conv3, Epoch 50

*Figure 13.* Comparison of gradient singular value spectra for methods with PAPO (solid lines) and without PAPO (dashed lines). Shown are Conv1 and Conv3 at epochs 20 and 50 for tasks T0, T3, T5, T7 and T9. In the legend, "Base" refers to the baselines without PAPO (i.e., ER, MAS, GPM).

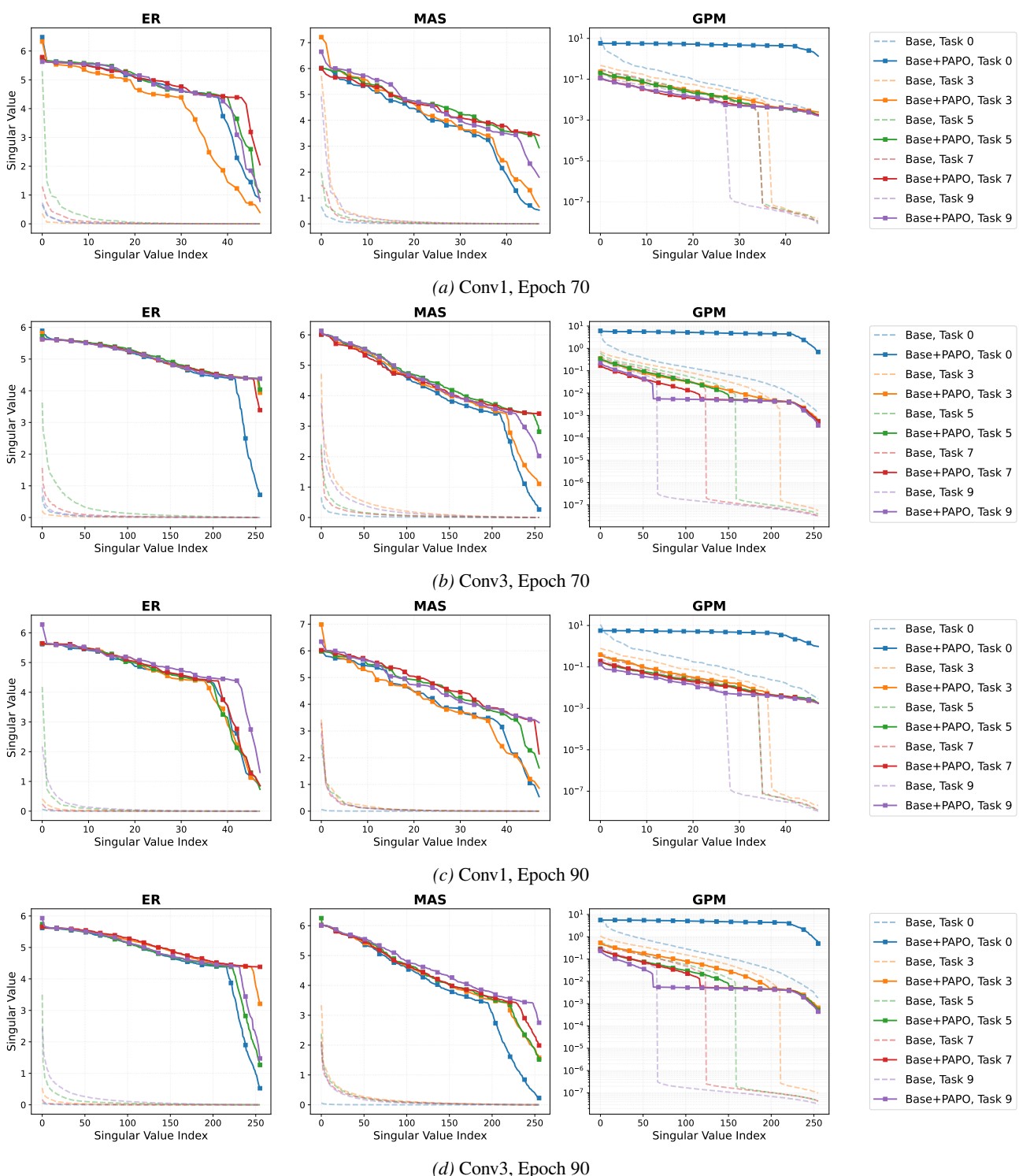

*(a)* Conv1, Epoch 70

*(b)* Conv3, Epoch 70

*(c)* Conv1, Epoch 90

*(d)* Conv3, Epoch 90

*Figure 14.* Comparison of gradient singular value spectra for methods with PAPO (solid lines) and without PAPO (dashed lines). Shown are Conv1 and Conv3 at epochs 70 and 90 for tasks T0, T3, T5, T7 and T9. In the legend, "Base" refers to the baselines without PAPO (i.e., ER, MAS, GPM).

## F.4. Computational Costs Introduced by PAPO

To evaluate the computational efficiency of PAPO, we run experiments on task 1572 (Samsum summary) from the SuperNI benchmark using LLaMA-2-7B. Two factors primarily affect PAPO's runtime: the PAPO application interval, which denotes the number of epochs between successive PAPO applications, and the number of Polar Express iterations, which denotes the iterations in Equation (6) used to approximate the polar factor of the gradient. As shown in Table 6, applying PAPO at every epoch increases runtime by 7.33% for InfLoRA and by 7.96% for GainLoRA relative to the baseline without PAPO. With a commonly used interval of 5, the runtime increases by 4.13% for InfLoRA and by 2.14% for GainLoRA. All experiments reported here were conducted on the same GPU and with identical hyperparameter settings. These results indicate that incorporating PAPO introduces only modest computational overhead in practice.

*Table 6.* Time cost for different PAPO application intervals (0, 1, 3, 5 and 10). The Polar Express approximation step is fixed at 5. An interval of 0 indicates that PAPO is not applied. Experiments were conducted on the same GPU and with identical hyperparameter settings.

| Interval | 0 | 1 | 3 | 5 | 10 |
|---|---|---|---|---|---|
| InfLoRA | 1622s | 1741s | 1705s | 1689s | 1632s |
| GainLoRA | 1959s | 2115s | 2046s | 2001s | 1988s |

*Table 7.* Time cost for different Polar Express iterations (i.e., 0, 1, 2, 3, 4, 5). The PAPO application interval is fixed at 5. An iteration of 0 indicates that PAPO is not applied. Experiments were conducted on the same GPU and with identical hyperparameter settings.

| Iterations | 0 | 1 | 2 | 3 | 4 | 5 |
|---|---|---|---|---|---|---|
| InfLoRA | 1622s | 1663s | 1675s | 1681s | 1686s | 1689s |
| GainLoRA | 1959s | 1977s | 1980s | 1979s | 1986s | 2001s |

## F.5. Experiments in the Non-Continual Learning Setting

We evaluate PAPO in a non-continual learning setting by training ResNet-18 on CIFAR-100. As shown in Table 8, PAPO consistently reduces the error rate.

*Table 8.* Top-1 and top-5 errors in the non-CL setting for ResNet-18 on CIFAR-100.

| Methods | Top1 error | Top5 error |
|---|---|---|
| SGD | 24.39 | 6.95 |
| +PAPO | 23.77 | 6.89 |

## F.6. Experiments on Online Continual Learning Benchmarks

We apply PAPO to online continual learning methods, including ER, OCM (Guo et al., 2022), and GSA (Guo et al., 2023), on the ImageNet100 benchmark. The hyperparameter settings of PAPO are listed in Table 9, and the other settings follow Wang et al. (2024a). The results are reported in Table 10. We find that PAPO also improves ACC and reduces FT in the online continual learning setting.

*Table 9.* Hyperparameter settings for Online CL Benchmarks.

| Methods | $\lambda$ | Interval | Scale factor |
|---|---|---|---|
| ER | 0.01 | 10 | 1 |
| OCM | 1 | 1 | 1 |
| GSA | 0.5 | 5 | 1 |

*Table 10.* ACC and FT on the ImageNet100 under online continual learning setting. Higher ACC and lower FT indicate better generalization and less forgetting.

| Methods | ACC↑ | FT↓ |
|---------|------|-----|
| ER      | 31.88 | 41.30 |
| +PAPO   | 36.40 | 34.33 |
| OCM     | 22.94 | 5.89 |
| +PAPO   | 33.90 | 4.09 |
| GSA     | 41.20 | 34.87 |
| +PAPO   | 43.28 | 33.94 |

## F.7. Other Experiments

In this section, we compare PAPO with several gradient correction-based methods, including isotropic perturbation, gradient rescaling, spectral regularization, SWA (Izmailov et al., 2018), and SAM (Foret et al., 2020).

For *isotropic perturbation*, we add isotropic Gaussian noise $\mathcal{G}$ to the gradient $G$ as $G = G + \lambda \frac{\|\mathrm{Polar}(G)\|_F}{\|\mathcal{G}\|_F + \epsilon}\mathcal{G}$, where the coefficient $\lambda$ is searched over $\{0.0001, 0.001, 0.01, 0.1, 1\}$. For *gradient rescaling*, we normalize the gradient to unit Frobenius norm and then rescale it to match the norm of the gradient produced by PAPO, namely $G = \frac{\|\mathrm{PAPO}(G)\|_F}{\|G\|_F + \epsilon}G$. We conduct these two experiments to show that the improvement brought by PAPO does not simply come from increasing the magnitude of the gradient.

We adopt *spectral regularization* from Lewandowski et al. (2025) and search over regularization coefficients in $\{0.0001, 0.001, 0.01, 0.1, 1\}$. For *SWA*, we use the implementation in `torch.optim.swa_utils` and search over both the starting epoch and the learning rate. We start SWA after $75\%$ of the total epochs and search over learning rates in $\{0.001, 0.01, 0.1\}$. For *SAM*, we follow the SAM setting in Yang et al. (2023a).

*Table 11.* Comparison with additional methods. Higher ACC and BWT indicate better generalization and reduced forgetting.

| Methods | ACC↑ | BWT↑ |
|---------|------|------|
| ER                        | $63.40 \pm 2.94$ | $-0.06 \pm 0.01$ |
| + Gaussian Noise          | $63.68 \pm 2.26$ | $-0.06 \pm 0.01$ |
| + Gradient Rescaling      | $63.59 \pm 1.04$ | $-0.05 \pm 0.01$ |
| + Spectral Regularization | $64.62 \pm 1.56$ | $-0.05 \pm 0.01$ |
| + SWA                     | $64.44 \pm 0.56$ | $-0.08 \pm 0.00$ |
| + SAM                     | $65.10 \pm 0.71$ | $-0.04 \pm 0.01$ |
| **+ PAPO**                | $\mathbf{72.41 \pm 1.17}$ | $\mathbf{-0.03 \pm 0.01}$ |
| MAS                       | $57.31 \pm 2.64$ | $-0.06 \pm 0.02$ |
| + Gaussian Noise          | $58.76 \pm 1.64$ | $-0.11 \pm 0.01$ |
| + Gradient Rescaling      | $60.75 \pm 0.99$ | $-0.03 \pm 0.01$ |
| + Spectral Regularization | $56.53 \pm 0.78$ | $-0.13 \pm 0.01$ |
| + SWA                     | $61.67 \pm 2.95$ | $-0.06 \pm 0.02$ |
| + SAM                     | $\mathbf{62.94 \pm 1.07}$ | $-0.08 \pm 0.01$ |
| **+ PAPO**                | $62.07 \pm 1.71$ | $\mathbf{-0.02 \pm 0.01}$ |
| GPM                       | $60.20 \pm 1.98$ | $-0.04 \pm 0.02$ |
| + Gaussian Noise          | $62.38 \pm 3.63$ | $-0.02 \pm 0.02$ |
| + Gradient Rescaling      | $61.11 \pm 1.46$ | $-0.02 \pm 0.01$ |
| + Spectral Regularization | $62.53 \pm 2.39$ | $-0.02 \pm 0.01$ |
| + SWA                     | $62.68 \pm 1.52$ | $-0.03 \pm 0.01$ |
| + SAM                     | $67.74 \pm 1.39$ | $-0.01 \pm 0.00$ |
| **+ PAPO**                | $\mathbf{69.10 \pm 1.48}$ | $\mathbf{+0.00 \pm 0.00}$ |

