# OpenReview forum: "Plasticity Activation via Polar Operator: A Plug-in Method for Balancing Stability and Plasticity"
_ICML.cc/2026/Conference — ICML 2026 regular_

### Official Review · Reviewer_EcLZ · 2026-03-05

**Soundness:** 3
**Presentation:** 3
**Significance:** 3
**Originality:** 4
**Overall Recommendation:** 5
**Confidence:** 4

**Summary:**

This work analyzes plasticity loss in continual learning and argues that several representative method familie show a recurring phenomenon: gradient singular value collapse. Essentially, updates concentrate into a small number of dominant directions while many other directions receive near-zero learning signal. To address this, the authors propose PAPO, a plug-in gradient modification that periodically adds a structured “polar” component to the gradient. Intuitively, this term aims to lift suppressed gradient directions in a near-uniform way while largely preserving the main update directions, with the goal of improving plasticity without destabilizing retention. The paper evaluates PAPO across multiple continual learning algorithms spanning the main families, on standard vision benchmarks, and additionally on continual instruction tuning with LoRA-style variants in an LLM setting. It includes ablations over key hyperparameters and approximation settings, and presents spectral diagnostics that support the collapse narrative and connect it to performance degradation over tasks.

**Compliance With Llm Reviewing Policy:**

Affirmed.

**Final Justification:**

This work represents a valuable contribution to the continual learning space, and the rebuttal reinforced this prior assessment. This paper should be accepted.

**Key Questions For Authors:**

-   If PAPO is compared against a compute-matched periodic isotropic gradient perturbation baseline (e.g., Gaussian or Rademacher noise with fairly tuned magnitude), does PAPO still outperform consistently?  (Weakness 1)
-	Is there a method-agnostic way to set the key hyperparameters (strength and application interval), for example via a normalization or schedule that transfers reliably across datasets and CL families without extensive tuning? Do you think gradient effective/stable rank (or a related collapse metric) could be used to inform or adapt these hyperparameters automatically?
-	Are there specific regimes where PAPO leads to performance degradation (e.g., particular method families such as projection-based approaches, certain strength ranges, or stages in the task sequence)?
-  Under substantially longer task horizons (e.g., 50–100 tasks), does PAPO keep the performance gains?

**Limitations:**

Yes.

**Strengths And Weaknesses:**

Strengths:

- Novel, structured polar updates. The core idea, using a polar-based gradient lift to counter collapse, is original and elegant.
- Simple, plug-and-play mechanism. PAPO is easy to integrate as a periodic add-on to many continual learning backbones, which strengthens practical appeal.
- The paper’s storyline is easy to follow and the diagnostic framing is well communicated.
-  The sensitivity studies (e.g., application frequency / approximation quality / strength) and the spectral plots support the central “collapse” hypothesis and help interpret when PAPO helps.

Weaknesses:
- A central concern for me is whether the gains come from the polar structure specifically or from generic spectral inflation. For instance you could look at (i) adding norm-matched isotropic noise (Gaussian or Rademacher) to the gradient, and (ii) simple magnitude-only interventions such as periodic gradient rescaling/renormalization (e.g., normalize the layer gradient to unit Frobenius norm and then rescale to the original norm, or clip-and-rescale). If PAPO consistently outperforms these, it would strongly support that the polar structure is essential.

-  Performance appears dependent on choosing the update strength and application interval. Overly aggressive settings can risk destabilizing learning (Figure 4 in the main body).

-	Section 3.1 figure explanation is scattered. The interpretation of the key spectral figure is interleaved with setup/definitions, which makes the takeaway harder to extract quickly. A tighter structure would improve readability.

---

> ### Author Rebuttal · Authors · 2026-03-31
>
> Thank you for your thoughtful suggestions and positive impressions of our work. We provide detailed responses to each point below.
>
> **Response to Weakness 1 & Question 1:** Thank you for the careful review and constructive comment. We conduct experiments on the MiniImageNet benchmark by ResNet-18 backbone. \
> (1) We add isotropic Gaussian noise $N$ to the gradient as $G = G + \lambda \frac{||\text{Polar}(G)||_F}{||N||_F + \epsilon} N$, where the coefficient $\lambda$ is searched over ${0.0001, 0.001, 0.01, 0.1, 1}$ as in PAPO, and $\text{Polar}(G)$ denotes the polar operator applied to $G$.\
> (2) As for gradient rescaling, normalizing the gradient to unit Frobenius norm and then rescaling it back to the original norm does not change the gradient. We assume you may refer to normalizing the gradient to unit Frobenius norm and then rescaling it to match the norm of the gradient obtained by PAPO: $G = \frac{||\text{PAPO}(G)||_F}{||G||_F + \epsilon} G.$ \
> As shown in the table, simply rescaling the original gradient improves model performance only slightly, and there remains a significant gap compared to PAPO. This highlights that the polar structure is essential.
>
> |Method|ACC|BWT|
> |-|-|-|
> |ER|63.40 ± 2.94|-0.06 ± 0.01|
> |+ Gaussian Noise|63.68±2.26|-0.06 ± 0.01|
> |+ Gradient Rescaling|63.59±1.04|-0.05 ± 0.01|
> |**+ PAPO**|**72.41 ± 1.17**|**-0.03 ± 0.01**|
> |MAS|57.31 ± 2.64|-0.06 ± 0.02|
> |+ Gaussian Noise|58.76 ± 1.64|-0.11 ± 0.01|
> |+ Gradient Rescaling|60.75 ± 0.99|-0.03 ± 0.01|
> |**+ PAPO**|**62.07 ± 1.71**|**-0.02 ± 0.01**|
> |GPM|60.20 ± 1.98|-0.04 ± 0.02|
> |+ Gaussian Noise|62.38 ± 3.63|-0.02 ± 0.02|
> |+ Gradient Rescaling|61.11 ± 1.46|-0.02 ± 0.01|
> |**+ PAPO**|**69.10 ± 1.48**|**+0.00 ± 0.00**|
>
> Furthermore, there are two key reasons why simple perturbations cannot replicate the effect of PAPO.
> (1) Adding random noise introduces singular directions that differ from those of the original gradient, which may alter the intrinsic properties of the gradient. In contrast, PAPO modifies the singular values without changing the original singular directions.\
> (2) Gradient rescaling, which merely multiplies the gradient by a scalar, does not meaningfully affect near-zero singular values. In fact, it may further enlarge the gap between the largest and smallest singular values, making the directions associated with small singular values even less effective. In contrast, PAPO modifies the singular values in an additive manner, which more effectively prevents them from approaching zero.
>
> **Response to Weakness 2:** Thank you for the careful review and constructive comment. The performance is related to the update strength $\lambda$. If $\lambda$ is too large, it may lead to a situation where the corrected singular values exhibit little variation. As a result, the singular vectors that originally played a dominant role in mitigating forgetting may no longer remain dominant, thereby losing the ability to mitigate forgetting. However, we observe that as long as the application interval is not too large, the performance drop is minimal. In practice, setting the interval to 5 is generally effective and efficient.
>
> **Response to Weakness 3:**  Thank you for the careful review and constructive comment. Following the suggestions from both you and Reviewer QHog, we will revise the caption of Figure 1 to read: “Gradient singular value spectra of a convolutional layer of AlexNet on CIFAR-100 for ER, MAS, and GPM across tasks.” All experimental setup details will be moved to Section 3.1 to improve clarity and readability.
>
> **Response to Question 2:** Thank you for this insightful comment. We have also considered this question before, but this approach seems infeasible for the following reasons.\
> (1) It is unclear  what effective or stable rank is beneficial for model performance, and such a choice would still require manual tuning. We believe that deriving an adaptive method would require a more rigorous theoretical analysis, such as characterizing the upper bounds of generalization, forgetting, and plasticity under different gradient ranks. We leave this challenging yet promising theoretical exploration for future work.\
> (2) Computing the effective or stable rank involves SVD, which introduces high computational overhead and slows down optimization. This would therefore contradict our goal of developing an efficient method.
>
> **Response to Question 3:** We observe that PAPO becomes ineffective when $\lambda$ is large in projection-based approaches, and the underlying reason is discussed in Response to Weakness 2. Therefore, a small $\lambda$ is preferred in projection-based approaches.
>
> **Response to Question 4:** This can be found in the Response to Weakness 1 for **Reviewer QHog**. We observe that PAPO can still improve plasticity over substantially longer task sequences.

---

> > ### Author Rebuttal · Reviewer_EcLZ · 2026-04-03
> >
> > I appreciate the authors’ detailed rebuttal. I have no further questions and I will maintain my score.

---

> > > ### Author Response · Authors · 2026-04-04
> > >
> > > Thank you very much for your positive and encouraging feedback. We truly appreciate your thoughtful review and support.

---

### Official Review · Reviewer_QHog · 2026-03-11

**Soundness:** 3
**Presentation:** 3
**Significance:** 4
**Originality:** 4
**Overall Recommendation:** 5
**Confidence:** 4

**Summary:**

The authors consider the stability-plasticity dilemma in continual learning settings where methods to mitigate catastrophic forgetting are already applied. They consider the problem through the lens of the gradient singular values. Directions where the singular value is high are safe to mitigate catastrophic forgetting; however, through continual learning, too many directions have close to zero singular value, causing a loss of plasticity. They propose an algorithm called PAPO that can be applied on top of several existing continual learning techniques. PAPO keeps the relative order of singular value (mitigating catastrophic forgetting), and uniformly increases close to zero singular value (improving plasticity).

**Compliance With Llm Reviewing Policy:**

Affirmed.

**Final Justification:**

I believed this is a good paper concerning the 4 dimensions evaluated, and the rebuttal reinforced this prior assessment. This paper should be accepted.

**Key Questions For Authors:**

I wonder why you did not choose a log-space plot in Figure 1 for ER and MAS just as you did for GPM?

**Limitations:**

Yes.

**Strengths And Weaknesses:**

Strenghts :

1. The paper is clear, easy to follow, and mathematically grounded.
2. PAPO improves consistently several continual learning techniques on several benchmarks of different natures, through rigorous and well-designed experiments, along with an ablation study and sensitivity analysis.
3. The paper goes beyond the toy benchmark, and shows that PAPO only marginally increases the computational cost on SOTA language model.

Weakness:

1. I believe the paper is lacking an experiment where the loss of plasticity of ER/MAS/GPM/TRGP is strikingly visible, supporting more clearly the claim that POPA will never lose plasticity (at some point, at the cost of forgetting inevitably). I suggest doing hundreds of MNIST permutations, where ER/MAS/GPM/TRGP should not be able to learn at all in the last permutation, because all the singular values have become nearly zero, PAPO should still be able to learn.

Minor comments:

1. The captions of Figure 1 and 2 should be more detailed. I had to look at the supplementary material to understand that Figure 1 shows the spectra for a convolution, for example.
2. I assume the convolutional layers are reshaped to 2D matrices for the singular value decomposition; this should be written somewhere.

---

> ### Author Rebuttal · Authors · 2026-03-31
>
> Thank you for your thoughtful suggestions and positive impressions of our work. We provide detailed responses to each point below.
>
> **Weakness:** I believe the paper is lacking an experiment where the loss of plasticity of ER/MAS/GPM/TRGP is strikingly visible, supporting more clearly the claim that POPA will never lose plasticity (at some point, at the cost of forgetting inevitably). I suggest doing hundreds of MNIST permutations, where ER/MAS/GPM/TRGP should not be able to learn at all in the last permutation, because all the singular values have become nearly zero, PAPO should still be able to learn.\
> **Response:** Thank you for the careful review and constructive comment. Following your recommendation, we conducted experiments on Permuted MNIST with 100 tasks (using 100 different permutations) and report the model performance on the final task. In particular, we observe that under GPM, the accuracy on the last task drops very quickly. This aligns with the discussion in Remark 3.2 of our paper, where we note that when the number of tasks is large, the singular value collapse phenomenon in the gradient matrix becomes more severe, leading to degraded model performance. In contrast, PAPO effectively corrects the singular values of the gradient, resulting in a significant improvement in accuracy on the final task.
> |Method|Last Accuracy|
> |-|-|
> |ER|95.2|-0.06±0.01|
> |ER + PAPO|**95.5**|
> |MAS|69.1|-0.06±0.02|
> |MAS + PAPO|**71.7**|
> |GPM|45.1|-0.04±0.02|
> |GPM + PAPO|**79.3**|
>
> **Minor comments 1:** The captions of Figure 1 and 2 should be more detailed. I had to look at the supplementary material to understand that Figure 1 shows the spectra for a convolution, for example.\
> **Response:** Thank you for the careful review and constructive comments. Following the suggestions from both you and Reviewer EcLZ, we will revise the caption of Figure 1 to read: “Gradient singular value spectra of a convolutional layer of AlexNet on CIFAR-100 for ER, MAS, and GPM across tasks.” In addition, all experimental setup details will be moved to Section 3.1 to improve clarity and readability.
>
> **Minor comments 2:** I assume the convolutional layers are reshaped to 2D matrices for the singular value decomposition; this should be written somewhere.\
> **Response:** Thank you for the careful review and constructive comment. For a convolutional layer, let $C_i$ and $C_o$ denote the number of input and output channels, respectively, and let $k$ be the kernel size. For any filter $W \in \mathbb{R}^{C_o \times C_i \times k \times k}$, we reshape it into a 2D matrix of size $(C_i k^2) \times C_o$. This allows us to perform singular value decomposition and analyze its spectral properties. This design is consistent with GPM[1], and we will clarify this preprocessing step in the Preliminary section.
>
> **Question:** I wonder why you did not choose a log-space plot in Figure 1 for ER and MAS just as you did for GPM?\
> **Response:** This is because we observe that the singular values in GPM exhibit an abrupt drop at a certain point, resulting in a cliff-like shape in the curve. To make this phenomenon more prominent in the figure, we use a log-space plot for GPM. However, this phenomenon does not appear in ER or MAS, so a log-space plot is not used for those methods.
>
> [1] Gradient Projection Memory for Continual Learning, ICLR 2021.

---

> > ### Author Rebuttal · Reviewer_QHog · 2026-04-03
> >
> > I appreciate the authors' rebuttal. The additional results clearly show that PAPO mitigates the loss of plasticity. I still believe this is a good paper and will maintain my score of 5 (Accept).

---

> > > ### Author Response · Authors · 2026-04-04
> > >
> > > Thank you very much for your positive and encouraging feedback. We truly appreciate your thoughtful review and support.

---

### Official Review · Reviewer_GLhr · 2026-03-13

**Soundness:** 3
**Presentation:** 3
**Significance:** 2
**Originality:** 2
**Overall Recommendation:** 4
**Confidence:** 4

**Summary:**

This paper studies the stability–plasticity trade-off in continual learning. The paper proposes PAPO, a plug-in that modifies the gradient by adding a polar-factor term, with the goal of lifting suppressed singular values while preserving the original singular vectors and the relative ordering of dominant directions.Empirically, the paper evaluates PAPO on several vision CL baselines (ER, MAS, GPM, TRGP) over PMNIST, Split CIFAR-100, MiniImageNet, and 5-Datasets, and on language CL baselines (InfLoRA and GainLoRA) on SuperNI with LLaMA-2-7B. The reported results show fairly consistent gains, with especially large improvements on MiniImageNet.

**Compliance With Llm Reviewing Policy:**

Affirmed.

**Final Justification:**

My primary concerns have been resolved, and I am pleased to raise my score. Please ensure that all new experimental results and clarifications are included in the camera-ready version.

**Key Questions For Authors:**

no

**Limitations:**

yes

**Strengths And Weaknesses:**

## Strengths

* The paper has a clear and easy-to-follow core idea. The motivation from gradient singular value collapse to a polar-operator-based correction is conceptually coherent, and the method itself is lightweight.

* The paper includes a theoretical argument for why a spectral cliff can arise in orthogonal-projection-based methods, and it also provides ablations.

* The empirical evaluation covers multiple CL families.


## Weaknesses

* The paper does not fully establish how continual-learning-specific the proposed benefit really is. The mechanism is essentially a spectrum-shaping modification of the gradient, and the experiments only compare CL baselines with and without PAPO. Without matched non-CL, sequential fine-tuning controls, or comparisons to more generic spectral/optimization corrections, it is hard to tell how much of the gain comes from something uniquely tied to continual learning rather than from a broader optimization effect.


* The vision experiments are on standard small-to-medium-scale CL benchmarks, and the language experiments are restricted to SuperNI with a single LLaMA-2-7B backbone. There is no ImageNet-scale class-incremental benchmark, no more realistic large-scale streaming setup, and no multimodal setting. That makes the generality of the empirical claim less convincing than the paper’s framing suggests.

* The argument assumes that the dominant singular directions largely correspond to forgetting mitigation while the suppressed tail mainly represents lost plasticity. That is intuitive, but the evidence remains mostly correlational. Include more direct diagnostics to show that the newly activated directions are indeed useful rather than simply larger. Besides, Singular value analysis for continual learning was already discussed in [1], and a similar analysis was later conducted in [2]. Please clarify the similarities and differences between your perspective and those prior works in the paper.

[1] Class-Incremental Learning via Dual Augmentation, NeurIPS 2021.

[2] Online Continual Learning through Mutual Information Maximization, ICML 2022.



* There is also no comparison against simpler alternatives that might explain part of the gain. Since PAPO is ultimately a gradient correction, the paper would be stronger if it compared against cheaper or more standard interventions such as generic gradient rescaling, isotropic perturbation, spectral regularization, or other optimizer-side corrections such as SWA, SAM for continual learning. Without such controls, it remains unclear whether the polar operator is essential or just one reasonable way to inject extra update directions.

---

> ### Author Rebuttal · Authors · 2026-03-31
>
> Thank you for your thorough review and constructive comments. We provide detailed responses to each point below.
>
> **Response to Weakness 1:** We evaluate PAPO in a non-CL setting by training ResNet-18 on CIFAR-100. As shown in the table below, PAPO consistently reduces the error rate. In fact, improving the accuracy of each task during continual learning training can lead to better plasticity. For language tasks, we apply LoRA for sequential fine tuning, and the results are presented in Table 2 of our paper. Comparisons with more generic spectral or optimization corrections are provided in the Response to Weakness 4.
>
> |Method|Top1 error|Top5 error|
> |-|-|-|
> |SGD|24.39|6.95|
> |**+ PAPO**|**23.77**|**6.89**|
>
> **Response to Weakness 2:** To simultaneously address your concerns regarding the ImageNet-scale class-incremental benchmark and the realistic large-scale streaming setup, we employ the ImageNet-100 benchmark to conduct streaming class-incremental tasks. We apply PAPO to online continual learning methods including ER, OCM [1], and GSA [2]. As shown in the table, applying PAPO to online continual learning methods consistently improves accuracy and reduces forgetting.  While time constraints prevented us from conducting multimodal experiments during the rebuttal period, it is important to note that PAPO operates at the optimization level. Consequently, its effectiveness is inherently agnostic to specific input modalities and model architectures. We believe that our comprehensive evaluations across continual vision tasks, continual language tasks, and streaming setups collectively underscore the generality of PAPO.
>
> |Method|Accuracy $\uparrow$|Forgetting $\downarrow$|
> |-|-|-|
> |ER|31.88|41.30|
> |**+ PAPO**|**36.40**|**34.33**|
> |OCM|22.94|5.89|
> |**+ PAPO**|**33.90**|**4.09**|
> |GSA|41.20|34.87|
> |**+ PAPO**|**43.28**|**33.94**|
>
> **Response to Weakness 3:** Thank you for the careful review and constructive comments. As shown in the Response to Weakness 1 & Question 1 for **Reviewer EcLZ**, we find that simply enlarging the gradient magnitude still leaves a significant gap compared to PAPO, indicating that activating directions indeed benefits model performance. Regarding the two papers you mentioned:
> [3] performs eigen-decomposition on the old and new feature extractors and defines the angle between the eigenvectors of the old and new tasks to capture the representation shift during incremental learning.
> [4] computes the eigenvalues and eigenvectors of the resulting representations to compare the holisticness of two different representation learning methods.
>
> In contrast, our work directly analyzes the singular values of the gradient, offering a more direct perspective on the optimization dynamics in continual learning.
>
> **Response to Weakness 4:** The results of generic gradient rescaling and isotropic perturbation can be found in the response to **Reviewer EcLZ**. For spectral regularization (Spec) and SWA, we search over the hyperparameters and report the best results in the table. Specifically, we adopt spectral regularization from [5] and search over regularization coefficients in $\{0.0001, 0.001, 0.01, 0.1, 1\}$. For SWA, we use the implementation in `torch.optim.swa_utils` and search over both the starting point and the learning rate. We start SWA after 75% of the total epochs and search over learning rates in $\{0.001, 0.01, 0.1\}$. As shown in the table, our method outperforms the others.
> |Method|ACC $\uparrow$|BWT $\uparrow$|
> |-|-|-|
> |ER|63.40 ± 2.94|-0.06 ± 0.01|
> |+ Spec|64.62 ± 1.56|-0.05 ± 0.01|
> |+ SWA|64.44 ± 0.56|-0.08 ± 0.00|
> |+ SAM|65.10 ± 0.71|-0.04 ± 0.01|
> |**+ PAPO**|**72.41 ± 1.17**|**-0.03 ± 0.01**|
> |MAS| 57.31 ± 2.64|-0.06 ± 0.02|
> |+ Spec|56.53 ± 0.78|-0.13 ± 0.01|
> |+ SWA|61.67 ± 2.95|-0.06 ± 0.02|
> |+ SAM|**62.94 ± 1.07**|-0.08 ± 0.01|
> |**+ PAPO**|62.07 ± 1.71|**-0.02 ± 0.01**|
> |GPM|60.20 ± 1.98|-0.04 ± 0.02|
> |+ Spec|62.53 ± 2.39|-0.02 ± 0.01|
> |+ SWA|62.68 ± 1.52|-0.03 ± 0.01|
> |+ SAM|67.74 ± 1.39|-0.01 ± 0.00|
> |**+ PAPO**|**69.10 ± 1.48**|**+0.00 ± 0.00**|
>
> [1] Online continual learning through mutual information maximization, ICML 2022.\
> [2] Dealing with cross-task class discrimination in online continual learning, CVPR 2023.\
> [3] Class-Incremental Learning via Dual Augmentation, NeurIPS 2021.\
> [4] Online Continual Learning through Mutual Information Maximization, ICML 2022.\
> [5] Learning Continually by Spectral Regularization, ICLR 2025.
>
> **Finally, if our responses have addressed your concerns to a satisfactory extent, we would be very grateful if you could consider updating your evaluation accordingly. We are also more than happy to clarify any remaining questions during the discussion period. We once again thank the reviewer for the constructive feedback and valuable suggestions on our work.**

---

> > ### Author Rebuttal · Reviewer_GLhr · 2026-04-04
> >
> > I thank the authors for their thorough response. My primary concerns have been resolved, and I am pleased to raise my score. Please ensure that all new experimental results and clarifications are included in the camera-ready version.

---

> > > ### Author Response · Authors · 2026-04-04
> > >
> > > Thank you for your thorough review and constructive comments. We will incorporate all new experimental results and clarifications into the camera-ready version.

---

### Official Review · Reviewer_hMWY · 2026-03-14

**Soundness:** 3
**Presentation:** 3
**Significance:** 3
**Originality:** 3
**Overall Recommendation:** 4
**Confidence:** 4

**Summary:**

The paper addresses the "Stability-Plasticity Dilemma" in Continual Learning (CL). The authors observe a phenomenon termed "singular value collapse" in the gradients of existing CL methods (such as ER, MAS, and GPM). This collapse restricts parameter updates to a small subset of directions, thereby suppressing model plasticity. To counter this, the authors propose PAPO, a plug-in method that utilizes a polar decomposition-based operator to modify the gradient's singular value spectrum. By uniformly enhancing small singular values while preserving the original singular vectors, PAPO activates suppressed gradient directions to boost plasticity without compromising stability. A computationally efficient approximation, "Polar Express," is introduced to make the method scalable. Experiments across CV and NLP benchmarks demonstrate that PAPO consistently improves the performance of various baseline CL algorithms.

**Compliance With Llm Reviewing Policy:**

Affirmed.

**Key Questions For Authors:**

PLZ see weakness and limitation.

**Limitations:**

1. The method relies on the assumption that the "suppressed" singular value directions are beneficial for new tasks. However, in some cases, these directions might contain noise that, if activated, could interfere with the preservation of old knowledge.
2. The introduction of $\lambda$ and the interval $s$ adds to the hyperparameter tuning burden, which can be cumbersome in real-world continual learning scenarios where the data distribution is unknown.

**Strengths And Weaknesses:**

Strengths,
1. The paper provides a empirical and theoretical motivation by identifying singular value collapse as a hidden bottleneck for plasticity in established CL methods.
2. As a plug-in method, PAPO demonstrates strong compatibility across different CL paradigms and different modalities.
3. The Polar Express iteration successfully mitigates the high computational cost typically associated with SVD.

Weaknesses,
1. The performance of PAPO is sensitive to the balance coefficient $\lambda$. If $\lambda$ is too large, the method can disrupt the stability of the model, leading to increased forgetting, especially in projection-based methods.
2. The application interval $s$ (how often PAPO is applied) is a critical hyperparameter that appears to be determined heuristically. A more principled way to trigger plasticity activation would strengthen the method.
3. While the paper identifies the collapse in various methods, it focuses primarily on the existence of the collapse rather than a deep theoretical explanation of why different methods (like ER vs. MAS) converge to similar collapsed spectra.
4. The related work section should be expanded to include and discuss other CL methods that also focus on gradient modification, like [1], [2] ,[3].

[1] Make Continual Learning Stronger via C-Flat, NeurIPS 2024

[2] A Faster Path to Continual Learning, CVPR 2026.

[3] ZeroFlow: Overcoming Catastrophic Forgetting is Easier than You Think, ICML 2025

---

> ### Author Rebuttal · Authors · 2026-03-31
>
> Thank you for your thorough review and constructive comments. We provide detailed responses to each point below.
>
> **Response to Weakness 1:** If $\lambda$ is too large, the dominant role of the directions that mitigate forgetting will no longer hold. When $\lambda$ is too large ($\lambda\gg \sigma$), the modified singular values become $\sigma+\lambda\approx\lambda$, causing different singular directions to have nearly equal weights. Consequently, the ability to mitigate forgetting is lost. Based on our empirical observations, due to the inherent properties of projection based methods, a smaller $\lambda$ is required, whereas other methods are more relatively robust to the choice of $\lambda$.
>
> **Response to Weakness 2:** We apply PAPO at an interval $s$ primarily to reduce computational cost. As shown in Figure 4, setting $s$ between 1 and 5 works well in most cases.  Moreover, when training LLaMA 2-7B on the SuperNI benchmark, we adopt the same setting and also observe improved model performance.
>
> **Response to Weakness 3:** We analyze the singular value collapse phenomenon by bounding the stable rank of the gradient and provide a proof sketch as follows. We simplify the discussion to a two-task scenario, and define the notation as follows. At the $t$-th training step of the $i$-th task, let $W_{i,t}$ and $G_{i,t}$ denote the weight matrix and gradient matrix, respectively, and let $w_{i,t}=\text{vec}(W_{i,t})$ and $g_{i,t}=\text{vec}(G_{i,t})$. Then the stable rank of $G_{2,t}$ can be written as
> $$\text{sr}(G\_{2,t})=\frac{||G\_{2,t}||\_F^2}{||G\_{2,t}||\_2^2}
> =\frac{||g\_{2,t}||\_2^2}{||G\_{2,t}||\_2^2}.$$
>
> (1) Following [1], $G\_{i,t}$ can be written as $G_{i,t}=\frac{1}{N}\sum_{j=1}^N(A_{i,j}-B_{i,j}W_{i,t}C_{i,j}).$
> For task 2, let $S_2=\frac{1}{N}\sum_{j=1}^N C_{2,j}\otimes B_{2,j}.$ Then $$g_{2,t}=(I-\eta S_2)g_{2,t-1}=(I-\eta S_2)^t g_{2,0}.$$
> However, in the continual learning setting, the initial gradient of the current task is not equal to the converged gradient of the previous task. Although the gradients differ, the current-task weight and the converged weight of the previous task are the same, i.e., $w_{2,0}=w_{1,T}$. Let $a_2=\frac{1}{N}\sum_{j=1}^N A_{2,j}$, we can further write
> $$g_{2,0}=a_2-S_2w_{1,T}
> =a_2-S_2\left(w_{1,0}+\eta\sum_{j=0}^{T-1}(I-\eta S_1)^j g_{1,0}\right).$$
>
> (2) Let $\lambda\_{2,1}<\lambda\_{2,2}$ be the two smallest distinct eigenvalues of $S_2$, and let $\mathcal{V}\_{2,1}$ be the corresponding minimal eigenspace. We decompose $g\_{2,0}=g\_{2,0}^{\parallel}+g\_{2,0}^{\perp}$, where $g\_{2,0}^{\parallel}\in\mathcal{V}\_{2,1}$ and $g\_{2,0}^{\perp}\perp\mathcal{V}\_{2,1}$. Then we have
> $$||G\_{2,t}||\_F^2=||g\_{2,t}||\_2^2\le(1-\eta\lambda\_{2,1})^{2t}||g\_{2,0}^{\parallel}||\_2^2+(1-\eta\lambda\_{2,2})^{2t}||g\_{2,0}^{\perp}||\_2^2,$$
> and
> $$||G\_{2,t}||\_2^2\ge (1-\eta\lambda\_{2,1})^{2t}||G\_{2,0}^{\parallel}||\_2^2.$$
>
> (3) Therefore,
> $$\mathrm{sr}(G\_{2,t})\le\mathrm{sr}(G\_{2,0}^{\parallel})+\left(\frac{1-\eta\lambda\_{2,2}}{1-\eta\lambda\_{2,1}}\right)^{2t}\frac{||g\_{2,0}^{\perp}||\_2^2}{||G\_{2,0}^{\parallel}||\_2^2}.$$
> Since $\lambda_{2,1}<\lambda_{2,2}$ and the step size satisfies $0<\eta<1/\lambda_{2,2}$, we have $\frac{1-\eta\lambda_{2,2}}{1-\eta\lambda_{2,1}}<1$. Therefore, the second term decays exponentially fast as $t$ increases. Consequently, the stable rank is eventually dominated by $\mathrm{sr}(G_{2,0}^{\parallel})$, which explains the singular value collapse phenomenon.
>
> **Response to Weakness 4:** Thank you for your thorough and constructive review. We will incorporate these methods into our related work section.
>
> **Response to Limitations 1:** As shown in the response to Weakness 1 & Question 1 for **Reviewer EcLZ**, adding noise to the gradient can sometimes be beneficial for continual learning. This can also be verified in [2], where perturbation improves plasticity. In fact, compared to simply adding random noise, our method has the advantage that it does not disrupt the original gradient direction. Adding random noise may change the original direction (for example, adding two vectors produces a new vector direction), which could potentially break the inherent properties of the algorithm.
>
> **Response to Limitations 2:** As stated in the responses to Weakness 1 and Weakness 2, the choice of $s$ can always be set between 1 and 5, depending on whether better performance or lower computation is prioritized. However, we are trying to propose an adaptive $\lambda$ without manual tuning. This requires a rigorous theoretical analysis of continual learning, such as characterizing the upper bounds of generalization, forgetting, and plasticity under different gradient ranks, which remains challenging in the field. We plan to address this challenge in future work.
>
> [1] GaLore: Memory-Efficient LLM Training by Gradient Low-Rank Projection, ICML 2024.\
> [2] Loss of plasticity in deep continual learning, Nature.

---

> > ### Author Rebuttal · Reviewer_hMWY · 2026-04-02
> >
> > Thank you for the additional analysis and the effort the authors put into the rebuttal. I have no further questions and will maintain my rating.

---

> > > ### Author Response · Authors · 2026-04-04
> > >
> > > Thank you very much for your positive and encouraging feedback. We truly appreciate your thoughtful review and support.

---

### Decision · Program_Chairs · 2026-04-30

**Decision:**

Accept (regular)

**Comment:**

All reviewers agreed that the submission should be accepted, after discussion with the authors. Therefore I recommend acceptance.